# SubseasonalClimateUSA: A Dataset for Subseasonal Forecasting and Benchmarking

**Soukayna Mouatadid[1], Paulo Orenstein[2], Genevieve Flaspohler[3], Miruna Oprescu[4], Judah Cohen[3], Franklyn Wang[5], Sean Knight[3], Maria Geogdzhayeva[3], Sam Levang[6], Ernest Fraenkel[3], Lester Mackey[4]**

[1]University of Toronto, [2]IMPA, [3]MIT, [4]Microsoft Research, [5]Harvard, [6]Salient Predictions

## Abstract

Subseasonal forecasting of the weather two to six weeks in advance is critical for resource allocation and advance disaster notice but poses many challenges for the forecasting community. At this forecast horizon, physics-based dynamical models have limited skill, and the targets for prediction depend in a complex manner on both local weather and global climate variables. Recently, machine learning methods have shown promise in advancing the state of the art but only at the cost of complex data curation, integrating expert knowledge with aggregation across multiple relevant data sources, file formats, and temporal and spatial resolutions. To streamline this process and accelerate future development, we introduce SubseasonalClimateUSA, a curated dataset for training and benchmarking subseasonal forecasting models in the United States. We use this dataset to benchmark a diverse suite of subseasonal models, including operational dynamical models, classical meteorological baselines, and ten state-of-the-art machine learning and deep learning-based methods from the literature. Overall, our benchmarks suggest simple and effective ways to extend the accuracy of current operational models. SubseasonalClimateUSA is regularly updated and accessible via the https://github.com/microsoft/subseasonal_data/ Python package.

## 1 Introduction

Weather and climate forecasting are fundamental scientific problems with many applications, including agriculture, energy grids, transportation and disaster prevention [36, 71, 55]. Indeed, short-term and long-term operational forecasts are critically employed in many sectors of our society and the world economy. However, skillful forecasts for the subseasonal regime—that is, 2 to 6 weeks ahead—still present operational challenges due to the chaotic nature of the weather [32] and to the interaction of weather and climate variables operating at different spatial and temporal scales [66].

In recent years, these challenges have spurred intense activity from both the meteorological and machine learning communities. On the one hand, steady advances have extended the reach of physics-based dynamical models of the atmosphere and oceans into the subseasonal realm [64, 50, 29]. On the other, parallel efforts from the machine learning community have led to improved predictive skill through new models trained on historical observational data and dynamical model forecasts [31, 8, 23, 2, 19, 78, 67–69, 60, 39].

Nevertheless, developing and benchmarking new subseasonal models remains challenging due to a lack of standardized, curated datasets targeting this forecast horizon. The data necessary for subseasonal predictions are often collected from multiple data sources, each with its own data processing pipeline, and then standardized into a common spatial and temporal resolution. Because there is uncertainty regarding the main drivers of subseasonal phenomena, domain experts are typically needed to determine which features are likely to carry signal, and different developers

37th Conference on Neural Information Processing Systems (NeurIPS 2023) Track on Datasets and Benchmarks.

employ different aggregation techniques over time and space. Finally, the changing nature of weather makes the forecasting task hard to compare across different regions and years. This, in turn, has spurred several subseasonal forecasting challenges to benchmark existing solutions [47, 46, 65].

In this paper, we introduce SubseasonalClimateUSA, a diverse collection of ground-truth measurements and dynamical forecasts for subseasonal prediction over the contiguous United States (U.S.). We include spatiotemporal measurements with known subseasonal impact (including, e.g., temperature, precipitation, sea surface temperature, and geopotential height); the states of known subseasonal drivers such as El Niño-Southern Oscillation (ENSO) and the Madden-Julian oscillation (MJO); and dynamical predictions for temperature and precipitation from eight operational models including the U.S. Climate Forecast System version 2 (CFSv2) and the leading subseasonal model from the European Centre for Medium-Range Weather Forecasts (ECMWF). The dataset is regularly updated and accessible via the open-source `subseasonal_data` Python package for easy retrieval.

We then use the SubseasonalClimateUSA data to benchmark a wide range of subseasonal models, highlighting their strengths and weaknesses. These models include traditional meteorological benchmarks (e.g., Persistence and Climatology), operational dynamical models (e.g., CFSv2 and ECMWF), and ten state-of-the-art deep learning (e.g., N-BEATS [48] and Informer [79]) and machine learning (e.g., CFSv2++ [39] and Prophet [62]) forecasters. Two of these models (Salient 2.0, the best-performing deep learning method, and LocalBoosting) were new creations of this work, and nine required new subseasonal forecasting implementations now available via the `subseasonal_toolkit` Python package. Model performance is measured for four standard subseasonal tasks: predicting average temperature 3-4 and 5-6 weeks ahead and predicting accumulated precipitation 3-4 and 5-6 weeks ahead. They are evaluated in terms of accuracy (measured by spatial root mean squared error) and skill (measured by uncentered anomaly correlation), over the years 2011–2020. Overall, we find that the simplest learned models typically outperform the meteorological baselines, the leading operational models, and the remaining learning methods. Additionally, we show that ensembling different methods through online learning leads to further gains in terms of both accuracy and skill.

Our aims in releasing SubseasonalClimateUSA are twofold. First, we aim to facilitate the development of skillful learning-based subseasonal forecasting models by providing a comprehensive, standardized, and machine-learning-friendly dataset. Second, we aim to provide standardized benchmarks for subseasonal forecasting progress. To this end, we define four core subseasonal prediction tasks that users can use as benchmarking targets: forecasting (i) temperature in weeks 3–4, (ii) temperature in weeks 5–6, (iii) precipitation in weeks 3–4, and (iv) precipitation in weeks 5–6. Significant advances in any of these tasks would have significant implications for the allocation of water resources, agricultural production, and disaster relief [47, 71].

**Related Work**     There are several datasets available for benchmarking weather models. For example, both the National Oceanic and Atmospheric Administration (NOAA) and the ECMWF provide reanalysis datasets tracking weather variables from the whole globe from the 1940s until today [24, 21], and global model simulations can be found in datasets provided by the World Climate Research Programme [13] and ECMWF [6]. More recently, several new datasets have been made available targeting specific AI applications in weather. For instance, classifying clouds [53], studying storm morphology [18] and nowcasting [16], predicting tropical cyclone intensity [35] and air quality metrics [4], and analyzing watershed-scale hydrometeorological time series [1] and river flows [17]. There are also more general-purpose datasets, such as WeatherBench [52], which provides a benchmark for forecasting different medium-range weather variables 3 to 5 days out. For a general overview of weather datasets for machine learning, see [11].

While these datasets have helped advance weather prediction in different tasks, there are no general datasets specifically targeting the subseasonal scale for the U.S. There have been instead several competitions targeting this lead time including the U.S. Bureau of Reclamation (USBR) Sub-Seasonal Climate Forecast Rodeos [47, 46] and the World Meteorological Organization Seasonal-to-Subseasonal (S2S) Artificial Intelligence (AI) Challenge [65]. Furthermore, the SubX Experiment [50] also makes a series of subseasonal models available for benchmarking (which are included in the SubseasonalClimateUSA dataset). Finally, the precursor of this work, the SubseasonalRodeo dataset of [23], targets only the Western U.S., offers only a static data snapshot ending in 2018, provides no forecasts from the leading subseasonal dynamical model (ECMWF), and includes only coarse-grained (monthly) forecasts from the North American Multi-Model Ensemble, with limited utility for weekly or biweekly forecasting.

In contrast, SubseasonalClimateUSA is a modern, regularly updated resource targeting the contiguous U.S. with granular (daily and subweekly) forecasts from ECMWF and seven other operational dynamical models in the SubX consortium [25]. Notably, both the present work and past studies have found complementary predictive signals in physics-based dynamical model forecasts and pure observational data that can lead to better forecasts than either data source alone [see, e.g., 23, 39]. In fact, recent work has demonstrated that even the least skillful operational dynamical models can produce forecasts with skill comparable to the best when corrected suitably with observational data [39]. As a result, we have endeavored to include both granular measurements and granular model forecasts in the SubseasonalClimateUSA dataset to best equip future model developers, researchers, and forecasters.

## 2   The SubseasonalClimateUSA dataset

The SubseasonalClimateUSA dataset houses a diverse collection of ground-truth measurements and dynamical model forecasts relevant to forecasting at subseasonal timescales. The dataset is regularly updated, CC BY 4.0 licensed, accessible via the open-source `subseasonal_data` Python package, and documented at the URL https://github.com/microsoft/subseasonal_data/blob/main/DATA.md. We summarize dataset contents, sources, and processing steps below and provide supplementary details in Appendix A.

**Data Collection and Processing**   Figure 1 summarizes the SubseasonalClimateUSA data collection and processing pipeline. The pipeline collects raw data from seven meteorological data sources (contributing different variables, resolutions, and file formats), passes all data through a common pre-processing pipeline, and outputs a standardized collection of machine-learning-ready Python Pandas DataFrames and Series objects stored in HDF5 format. Each file contributes data variables falling into one of three categories: (i) spatial (varying with the target grid point but not the target date); (ii) temporal (varying with the target date but not the target grid point); (iii) spatiotemporal (varying with both the target grid point and the target date). Data representing ground-truth measurements are typically downloaded daily on $0.25°$ or $0.5°$ latitude-longitude grids. However, subseasonal forecasts are typically issued on coarser $1°$ or $1.5°$ grids and averaged over two-week periods [47, 46, 65]. As a result, unless otherwise noted below, temporal and spatiotemporal variables arising from daily data sources were derived by averaging input values over a 14-day rolling window, and spatial and spatiotemporal variables were derived by interpolating input data to a $1°$ latitude-longitude grid and retaining only the grid points belonging to the contiguous U.S. To accommodate ECMWF forecasts, which were only made available on a $1.5°$ latitude-longitude grid [12], we additionally download SubX forecasts at $1.5°$ resolution and interpolate temperature and precipitation onto the same grid.

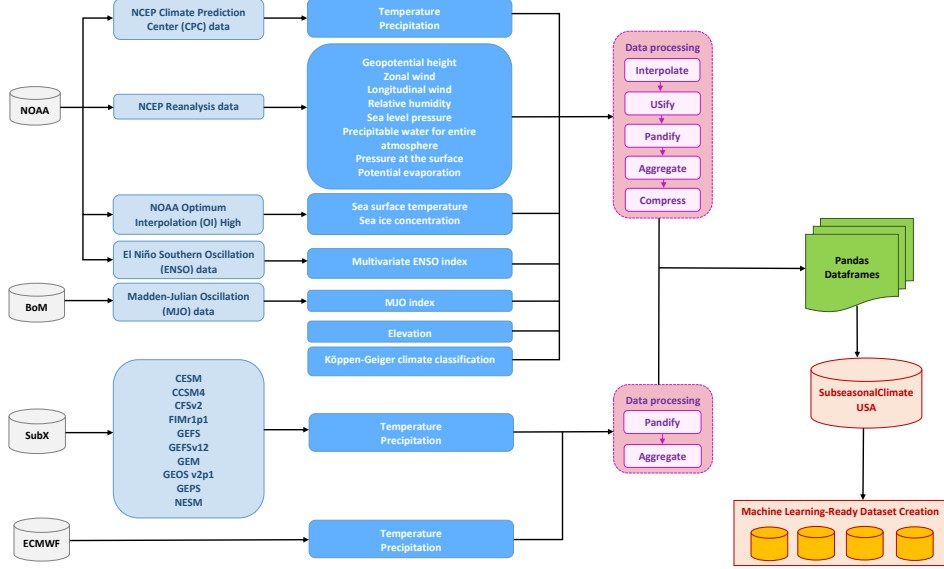

Figure 1: Schematic of the SubseasonalClimateUSA data collection and processing pipeline.

Following the protocol adopted by the USBR Sub-Seasonal Climate Forecast Rodeos [47, 46], our temperature and precipitation variables are interpolated onto fixed $1° \times 1°$ (`NUM_LAT=181`, `NUM_LON=360`) and $1.5° \times 1.5°$ (`NUM_LAT=121`, `NUM_LON=240`) grids using the NCAR Command Language function `area_hi2lores_Wrap` with arguments `new_lat = latGlobeF(NUM_LAT, ''lat'', ''latitude'', ''degrees_north'')`; `new_lon = lonGlobeF(NUM_LON, ''lon'', ''longitude'', ''degrees_east'')`; `wgt = cos(lat*pi/180.0)` (so that points are weighted by the cosine of the latitude in radians); `opt@critpc = 50` (to require only 50% of the values to be present to interpolate); and `fiCyclic = True` (indicating global data with longitude values that do not quite wrap around the globe). All remaining spatial and spatiotemporal variables are interpolated using the Climate Data Operators operator `remapdis` (distance-weighted average interpolation) with target grid `r360x181`.

**Data Features**   The variables comprising the SubseasonalClimateUSA dataset include:

- **Temperature** (global and U.S., 1979–present): daily mean of maximum and minimum temperature at 2 meters in °C [14, 40].

- **Precipitation** (global and U.S., 1948–present): daily accumulated precipitation in mm, aggregated by summing over a rolling two-week window instead of averaging [77, 7, 76, 41, 42].

- **Sea surface temperature and sea ice concentration** (global, 1981–present): daily variables that track variability in the oceans; the top three principal components for each variable were extracted using global 1981–2010 loadings [54, 45].

- **Stratospheric geopotential height, zonal winds, and longitudinal winds** (global, 1948–present): daily geopotential height at 10, 100, 500, 850 millibars and zonal and longitudinal winds at 250 and 925 millibars as indicators of polar vortex variability; the top three principal components of each feature were extracted from global 1948–2010 loadings [24, 44].

- **Surface pressure and relative humidity** (U.S., 1948–present): daily pressure and relative humidity near the surface (sigma level 0.995) [24, 44].

- **Sea level pressure, precipitable water for entire atmosphere, and potential evaporation** (U.S., 1948–present): daily mean of pressure in millibars, amount of water in the atmosphere available for precipitation in kg/m$^2$, and potential evaporation rate at surface [24, 44].

- **Elevation** and **Köppen-Geiger climate classification** (global): multi-resolution terrain elevation data [9] and Köppen-Geiger climate classification [26] for each grid point.

- **Madden-Julian Oscillation** (MJO, 1974–present): daily measure of tropical convection known to impact subseasonal climate; phase and amplitude were extracted (but not aggregated) [70, 38].

- **Multivariate ENSO index** (MEI.v2, 1979–present): bimonthly scalar summary of the state of the El Niño–Southern Oscillation, an ocean-atmosphere coupled climate mode [73–75, 43].

- **CFSv2** (U.S., 1999–present): daily 32-member ensemble mean forecasts of temperature and precipitation from the coupled atmosphere-ocean-land dynamical model with 0.5-29.5 day lead times [57, 25, 61].

- **SubX** (U.S., 1999–present): subweekly forecasts and hindcasts from seven dynamical models (GMAO-GEOS, NRL-NESM, RSMAS-CCSM4, ESRL-FIM, EMC-GEFS, ECCC-GEM, NCEP-CFSv2) and their multi-model mean for temperature and precipitation on a $1.5° \times 1.5°$ latitude-longitude grid [25, 61].

- **ECMWF** (U.S., 1995–present): control and perturbed forecasts and reforecasts of precipitation and temperature on a $1.5° \times 1.5°$ latitude-longitude grid [64, 12].

An example of SubseasonalClimateUSA observations and dynamical model forecasts is displayed in Figure 2.

**Dataset Limitations**   Here, we highlight several limitations of the SubseasonalClimateUSA dataset. First, the dataset was designed for forecasting in the contiguous U.S., and hence several variables are only available in that region. In future work, we aim to develop an analogous dataset for global subseasonal forecasting. Second, subseasonal forecasts are commonly made at the biweekly temporal resolution and $1°$ or $1.5°$ spatial resolution provided in the SubseasonalClimateUSA dataset [47, 46, 65]; however, these resolutions alone are insufficient for more localized forecasting problems without additional downscaling. Finally, many of our variables have undergone regridding via interpolation, which, while standard, can still introduce inaccuracies.

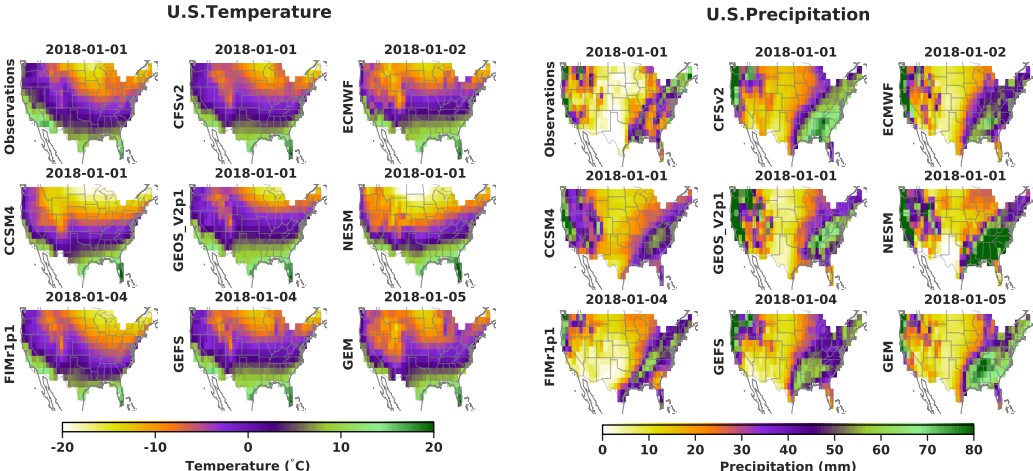

Figure 2: Example of SubseasonalClimateUSA observations and dynamical model forecasts.

## 3   Subseasonal Forecasting Tasks

We study model performance through four canonical subseasonal forecasting tasks: predicting two variables—average temperature (°C) and accumulated precipitation (mm) over a two-week period—each over two time horizons: 15–28 days ahead (weeks 3–4) and 29–42 days ahead (weeks 5–6). We forecast each variable at $G = 862$ locations on a $1° \times 1°$ latitude-longitude grid covering the contiguous U.S. These prediction targets and time horizons were the focus of the Sub-Seasonal Climate Forecast Rodeos [47, 46], two yearlong real-time forecasting competitions sponsored by USBR and NOAA to advance the state of subseasonal climate prediction. The same targets are used by water managers to apportion water resources, control wildfires, and anticipate droughts and other extreme weather [47, 71].

We evaluate each forecast according to two metrics recommended by the USBR [47, 46]: root mean squared error (RMSE) and *skill* (also known as uncentered anomaly correlation [72]). For a two-week period starting on date $t$, let $\mathbf{y}_t \in \mathbb{R}^G$ denote the vector of ground-truth measurements $y_{t,g}$ for each grid point $g$ and $\hat{\mathbf{y}}_t \in \mathbb{R}^G$ denote a corresponding vector of forecasts. In addition, define climatology $\mathbf{c}_t$ as the average ground-truth values for a given month and day over the years 1981-2010. Then the RMSE is given by

$$\text{RMSE}(\hat{\mathbf{y}}_t, \mathbf{y}_t) = \sqrt{\tfrac{1}{G} \sum_{g=1}^{G} (\hat{y}_{t,g} - y_{t,g})^2} \in \mathbb{R}_+$$

with a smaller value indicating a more accurate forecast, and skill is defined by

$$\text{skill}(\hat{\mathbf{y}}_t, \mathbf{y}_t) = \tfrac{\langle \hat{\mathbf{y}}_t - \mathbf{c}_t, \mathbf{y}_t - \mathbf{c}_t \rangle}{\|\hat{\mathbf{y}}_t - \mathbf{c}_t\|_2 \cdot \|\mathbf{y}_t - \mathbf{c}_T\|_2} \in [-1, 1]$$

with a larger value indicating higher quality. For a collection of dates, we report average RMSE and average percentage skill, which is 100 times the average skill.

**Training and Validation Recommendations**   We recommend adopting the Wednesdays of each complete year from 2011 onward as a standard test set when evaluating performance year by year (as in Figure 3) and the Wednesdays from the ten year period 2011–2020 as a standard test set when comparing with the overall (Table 1) or seasonal (Figure 3) performance reported in this work. We also recommend a progressive training and validation protocol in which, to produce a forecast for a given target date, a model can be (re)trained and (re)tuned using any data fully observable on the associated forecast issuance date (i.e., 14 days prior for weeks 3–4 and 28 days prior for weeks 5–6). All of the models evaluated in Section 5 respect this protocol. In addition, the SubseasonalClimateUSA dataset provides convenient combination dataframes containing target variables associated with predictive features lagged by an appropriate amount to ensure that each predictive feature was fully observed on the issuance date associated with the target date.

# 4 Benchmark Models

Our experiments will evaluate three classes of forecasting methods: three standard meteorological baselines, the adaptive bias correction (ABC) models introduced in [39], and seven other state-of-the-art machine learning and deep learning methods drawn from the literature. We will also evaluate ensemble forecasts derived from these models. Open-source model implementations are available via the `subseasonal_toolkit` Python package. Most models train on all available data (subject to the progressive training and validation constraint of Section 3) and tune hyperparameters using a second round of progressive evaluation over the prior three years. Appendix B contains supplementary implementation details for each model, including training, hyperparameter tuning, and testing details.

**Meteorological Baselines**    We first consider three standard subseasonal forecasting baselines.

CLIMATOLOGY.  Climatology is a standard subseasonal benchmark for the expected temperature or precipitation at a location. For a given grid point and target date, it forecasts the average value of the target variable on the same day and month over 1981-2010 [3].

DEBIASED CFSV2.  CFSv2 is the U.S. operational dynamical model commonly used for subseasonal forecasting [57]. Debiased CFSv2 is a corrected ensemble forecast used as a benchmark in the two Subseasonal Climate Forecast Rodeo competitions [46, 23]. First, a CFSv2 ensemble forecast is formed by averaging 32 forecasts for the target period based on 4 different model initializations produced at 8 different lead times. The ensemble is then *debiased* by adding the mean value of the target variable on the target month and day over the period 1999-2010 and subtracting the mean ensemble CFSv2 reforecast over the same period.

PERSISTENCE.  This baseline [37, 69] forecasts the most recently observed two-week target value.

**ABC Models**    We next evaluate the ABC models introduced in [39]. ABC is a hybrid physics-plus-learning approach that takes as input a dynamical model forecast (here, CFSv2) and uses the historical record of prior forecasts and observations to correct the model's output and improve predictive skill. While the three learning models contributing to ABC described below are simple and computationally inexpensive, Section 5 shows that each enhancement improves over both operational practice and state-of-the-art learning techniques.

CLIMATOLOGY++.  Climatology++ is as an adaptive form of Climatology that learns how many prior years and how many dates in a window around the target date to include in a smoothed historical mean or geometric median estimate for a given grid point, target date, and target variable; importantly, unlike a static climatology, Climatology++ allows these learned window sizes to vary over time to adapt to noise levels and variability.

CFSV2++.  CFSv2++ is a learned correction for raw CFSv2 forecasts. After averaging CFSv2 forecasts over a range of issuance dates and lead times, CFSv2++ debiases the ensemble forecast by adding the mean value of the target variable and subtracting the mean forecast over a learned window of observations around the target day of year. The range of ensembled lead times, the number of averaged issuance dates, and the size of the observation window employed are selected adaptively.

PERSISTENCE++.  For each grid point and target date, Persistence++ predicts a target variable as a function of lagged measurements, Climatology, and CFSv2 forecasts observable for the same grid point on the forecast issuance date. These features are combined using a linear least squares regression trained on all available historical data available as of the forecast issuance date.

**State-of-the-art Learning Methods**    We also consider seven state-of-the-art learning methods.

AUTOKNN.  AutoKNN [23] was part of a winning solution in the Subseasonal Climate Forecast Rodeo I [46]. Our implementation adapts it to target RMSE as an error metric.

INFORMER.  The Informer [79] is a transformer-based deep learning model for time series shown to have state-of-the-art performance on a number of short term weather forecasting tasks.

LOCALBOOSTING.  LocalBoosting is a decision tree model using CatBoost [51] over features in a bounding box around the target to extract meaningful spatial information.

MULTILLR.  The MultiLLR [23] model is a customized backward stepwise procedure to select SubseasonalClimateUSA features relevant for prediction and local linear regression to combine those features into a forecast for each grid point.

N-BEATS. N-BEATS [48] is a neural network time series forecaster that obtained state-of-the-art results on the Makridakis M3 [33] and M4 [34] benchmarks for time-series forecasting.

PROPHET. The Prophet model of [62] is an additive regression model for time-series and a winning solution in the Subseasonal Forecast Rodeo II [46].

SALIENT 2.0. Salient 2.0 is an ensemble of fully-connected neural networks, trained on historical sea surface temperature (SST) data. It is based on Salient [59], a winning solution for the Subseasonal Forecast Rodeo I [46].

**Ensembles**    Ensemble forecasts that combine the predictions of multiple models have been shown to improve the performance of long-, mid-, and short-range operational forecasting [10, 49, 23]. Here, we evaluate two ensembling strategies: Uniform ABC, which forms an equal-weighted average of the ABC model forecasts [27], and Online ABC, which uses the AdaHedgeD algorithm of [15] to choose weights adaptively to reflect relative model performance. See Appendices B.11 and B.12 for details.

## 5    Benchmark Results

We now turn to evaluating the models of Section 4 on the four subseasonal forecasting tasks of Section 3. We generate forecasts for each Wednesday in the years 2011–2020 and, for each reported period, we assess both mean RMSE relative to a baseline model and average percentage skill.

**Overall Performance**    Table 1 summarizes model performance across the entire ten-year period 2011–2020. On each task, we find that the ABC models provide both the best RMSE and the best skill performance. For example, on the two precipitation tasks, Climatology++ alone improves upon debiased CFSv2 RMSE by 9% and skill by 161-250%, outperforming each of the meteorological baselines and state-of-the-art learning methods. On the two temperature tasks, CFSv2++ and Persistence++ each outperform all meteorological baselines and state-of-the-art learning methods, with CFSv2++ improving debiased CFSv2 RMSE by 6-7% and skill by 30-53%. On every task, we observe further improvements in both RMSE and skill by ensembling the predictions of the three ABC models.

One might wonder how the simple ABC models are able to outperform both the standard meteorological baselines and the state-of-the-art learning methods. We believe the answer to this question is multifaceted. First, even the leading physics-based dynamical models are subject to inaccuracies due to inexact measurement, incomplete representation of the environment, imperfect simulation, and chaos [32]. Second, many of the more elaborate machine learning models studied in this work appear to be prone to overfitting in the presence of the relatively high noise levels of subseasonal forecasting. Third, the best-performing models, while relatively simple, are hybrid physics-plus-learning models designed to leverage the strengths of an underlying dynamical model while simultaneously enhancing its predictive skill by reducing its systemic bias.

Amongst the state-of-the-art learning methods, we find that Prophet performs the best for temperature weeks 5-6 and the two precipitation tasks, while MultiLLR performs the best for temperature weeks 3-4. Amongst the neural network methods (Informer, N-BEATS, and Salient 2.0), Salient 2.0 is the top performer with skill that rivals the other learning methods and precipitation RMSE that outpaces debiased CFSv2. In the more detailed analyses to follow, we omit Informer and N-BEATS due to space constraints and their relatively poor performance overall.

**Performance by Season and by Year**    We observe the same trends when performance is disaggregated by season or by year (Figure 3). For example, in every season, Climatology++ outperforms debiased CFSv2 and each state-of-the-art learner for the two precipitation tasks, while CFSv2++ and Persistence++ outperform debiased CFSv2 and each state-of-the-art learner each season for the two temperature tasks. Similarly and despite significant heterogeneity in all models' performances from year to year, the ABC models provide the best RMSE performance in 9 out of 10 years for temperature weeks 3-4 and in 10 out of 10 years for the two precipitation tasks. Indeed, Persistence++ alone dominates the temperature weeks 3-4 baselines and learners every year save 2019, and the more detailed RMSE summary of Appendix C.2 shows that the Uniform and Online ABC ensembles dominate the precipitation baselines and learners every year. In Figure 3, we observe nearly identical improvement patterns for skill.

Table 1: Average percentage skill and percentage improvement over mean debiased CFSv2 RMSE across 2011–2020 in the contiguous U.S. along with a 95% bootstrap confidence interval. The best performing model in each model group is bolded, and the best performing model overall is shown in green.

| | | % Improvement over Mean Deb. CFSv2 RMSE | | | | Average % Skill | | | |
| | | Temperature | | Precipitation | | Temperature | | Precipitation | |
| Group | Model | Weeks 3-4 | Weeks 5-6 | Weeks 3-4 | Weeks 5-6 | Weeks 3-4 | Weeks 5-6 | Weeks 3-4 | Weeks 5-6 |
|---|---|---|---|---|---|---|---|---|---|
| Baselines | Climatology | **0.13±1.33** | **2.93±1.23** | 7.79±0.55 | 7.51±0.48 | – | – | – | – |
| | Deb. CFSv2 | – | – | – | – | **24.94±1.88** | **19.12±1.94** | 5.77±1.11 | 4.28±1.09 |
| | Persistence | −109.94±5.3 | −170.1±7.56 | −28.27±1.47 | −31.92±1.53 | 10.64±2.26 | 6.22±2.38 | **8.31±0.99** | **7.41±1.01** |
| ABC | Climatology++ | 2.06±1.35 | 4.83±1.18 | **8.86±0.53** | **8.57±0.5** | 18.61±1.95 | 18.87±1.92 | 15.04±1.02 | 14.99±1.03 |
| | CFSv2++ | 5.94±1.08 | **7.09±1.02** | 8.37±0.5 | 8.06±0.45 | 32.38±1.75 | **29.19±1.76** | **16.34±1.03** | **16.09±1.07** |
| | Persistence++ | **6.00±1.06** | 6.43±0.99 | 8.61±0.51 | 7.89±0.45 | **32.4±1.71** | 26.73±1.67 | 13.38±0.91 | 9.77±0.9 |
| Learning | AutoKNN | 0.93±1.33 | 3.22±1.25 | 7.73±0.56 | 7.33±0.49 | 12.43±1.67 | 8.56±1.52 | 6.66±1.23 | 5.93±1.33 |
| | Informer | −40.61±4.4 | −39.57±3.89 | −2.05±0.86 | −2.53±0.83 | 0.55±2.22 | 0.01±2.20 | 6.15±1.28 | 5.86±1.31 |
| | LocalBoosting | −0.76±1.24 | −0.29±1.24 | 7.36±0.58 | 6.89±0.5 | 14.44±1.59 | 12.69±1.64 | 10.82±0.87 | 9.72±0.86 |
| | MultiLLR | **2.45±1.18** | 2.21±1.24 | 7.12±0.51 | 6.65±0.47 | **24.5±1.77** | 16.68±1.85 | 9.49±1.03 | 7.97±1.04 |
| | N-Beats | −46.71±2.48 | −52.05±2.89 | −19.19±0.92 | −21.32±0.89 | 9.21±1.40 | 4.16±1.39 | 5.48±0.59 | 4.46±0.62 |
| | Prophet | 1.13±1.4 | **3.78±1.26** | **8.42±0.55** | **8.12±0.51** | 20.21±1.54 | **19.78±1.57** | **13.51±0.87** | **13.41±0.89** |
| | Salient 2.0 | −6.95±1.69 | −4.05±1.73 | 2.97±0.74 | 2.65±0.69 | 11.24±2.04 | 11.77±2.03 | 10.11±1.36 | 9.99±1.31 |
| Ensembles | Uniform ABC | 6.47±1.09 | 7.55±0.99 | 9.47±0.5 | **9.05±0.45** | **33.58±1.8** | **30.56±1.7** | **18.94±0.98** | **18.35±1.01** |
| | Online ABC | **6.67±0.99** | **7.67±0.98** | **9.51±0.51** | 9.04±0.42 | 33.27±1.71 | 30.06±1.72 | 18.86±1.01 | 17.91±1.01 |

**Spatial Performance** Figure 4 displays how the errors of the leading models are distributed across the contiguous U.S. Here we focus on the ABC models, the best deep learning model (Salient 2.0), the best learning model (Prophet), and the best ensemble model (Online ABC) and provide RMSE improvement maps for the remaining models in Appendix C.4. At each grid point location, darker green indicates stronger improvement over debiased CFSv2, and we simultaneously witness two noteworthy phenomena. First, the improvements of each model are heterogeneous across space with the strongest improvements often occurring in the Western U.S., in Florida, or in Maine. Second, despite this heterogeneity, the ABC models consistently outperform the state-of-the-art learners.

**ECMWF Comparison** To compare the ABC models with the state-of-the-art ECMWF S2S dynamical model, we evaluate on the $1.5° \times 1.5°$ grid and 2016-2020 twice-weekly target date range available from [12, 66]. We debias both the ECMWF control forecast and its 50-member ensemble forecast following the operational protocol described by [69]; see Appendix B.13 for more details. Table 2 summarizes model performance. Remarkably, for precipitation, Climatology++ improves upon both the skill and the RMSE of ECMWF, despite making no use of dynamical model forecasts. Meanwhile, the Uniform ABC ensemble outperforms ECMWF in both metrics for all four tasks.

Table 2: Average percentage skill and percentage improvement over mean debiased CFSv2 RMSE across 2016-2020 in the contiguous U.S. along with a 95% bootstrap confidence interval. The best performing model in each model group is bolded, and the best overall is shown in green.

| | | % Improvement over Mean Deb. CFSv2 RMSE | | | | Average % Skill | | | |
| | | Temperature | | Precipitation | | Temperature | | Precipitation | |
| Group | Model | Weeks 3-4 | Weeks 5-6 | Weeks 3-4 | Weeks 5-6 | Weeks 3-4 | Weeks 5-6 | Weeks 3-4 | Weeks 5-6 |
|---|---|---|---|---|---|---|---|---|---|
| Baselines | Climatology | **1.56±1.39** | **3.92±1.27** | 8.7±0.5 | 7.56±0.53 | – | – | – | – |
| | Deb. CFSv2 | – | – | – | – | **22.64±2.01** | **15.71±2.09** | 2.84±1.16 | 1.68±1.11 |
| | Persistence | −105.57±5.58 | −169.22±7.39 | −28.05±1.56 | −33.43±1.65 | 9.12±2.48 | 2.27±2.48 | **8.11±1.07** | **6.21±1.04** |
| ABC | Climatology++ | 3.88±1.38 | 6.44±1.13 | **9.79±0.53** | **8.61±0.51** | 22.09±1.89 | 23.2±1.91 | **15.34±1.05** | **15.06±1.07** |
| | CFSv2++ | 5.65±1.04 | 6.65±0.96 | 8.94±0.51 | 7.6±0.46 | 30.91±1.8 | 26.87±1.93 | 14.6±1.23 | 13.85±1.2 |
| | Persistence++ | **7.06±1.05** | **7.86±0.95** | 9.06±0.48 | 7.57±0.46 | **31.46±1.92** | **28.04±1.87** | 10.03±0.99 | 6.61±0.95 |
| ECMWF | Debiased Control | −29.05±2.39 | −33.25±2.69 | −30.81±1.45 | −31.84±1.43 | 18.52±1.82 | 13.71±1.85 | 0.82±1.07 | 3.17±1.08 |
| | Debiased Ensemble | **4.62±1.19** | **3.69±1.27** | **7.90±0.5** | **6.41±0.46** | **32.27±1.69** | **26.61±1.71** | **13.12±1.16** | **9.10±1.09** |
| Ensembles | Uniform ABC | **7.43±1.05** | **8.27±0.94** | 10.04±0.5 | **8.77±0.46** | **32.77±1.79** | **29.75±1.87** | 16.53±1.15 | **15.71±1.17** |
| | Online ABC | 7.2±1.07 | 7.96±0.98 | **10.08±0.48** | 8.62±0.47 | 32.22±1.82 | 28.38±1.87 | **17.19±1.12** | 15.42±1.15 |

**Graphcast Comparison** Recently, the GraphCast deep learning model [28] was shown to outperform both the leading dynamical model and the Pangu-Weather deep learning model [5] on a range of 0 to 10-day weather forecasting tasks. While GraphCast was not developed for subseasonal forecasting, we can, as recommended by an anonymous reviewer, benchmark its performance on our subseasonal forecasting tasks. For this evaluation, we restrict our standard test set to the years 2018–2020, as GraphCast was trained on data through 2017, and report performance and additional experimental details in Appendix C.6. Consistent with the other deep learning methods benchmarked in Table 1, GraphCast outperforms debiased CFSv2 in terms of skill, underperforms debiased CFSv2 in terms of RMSE, and strongly underperforms the ABC ensemble models in both metrics when forecasting either temperature or precipitation in weeks 3-4.

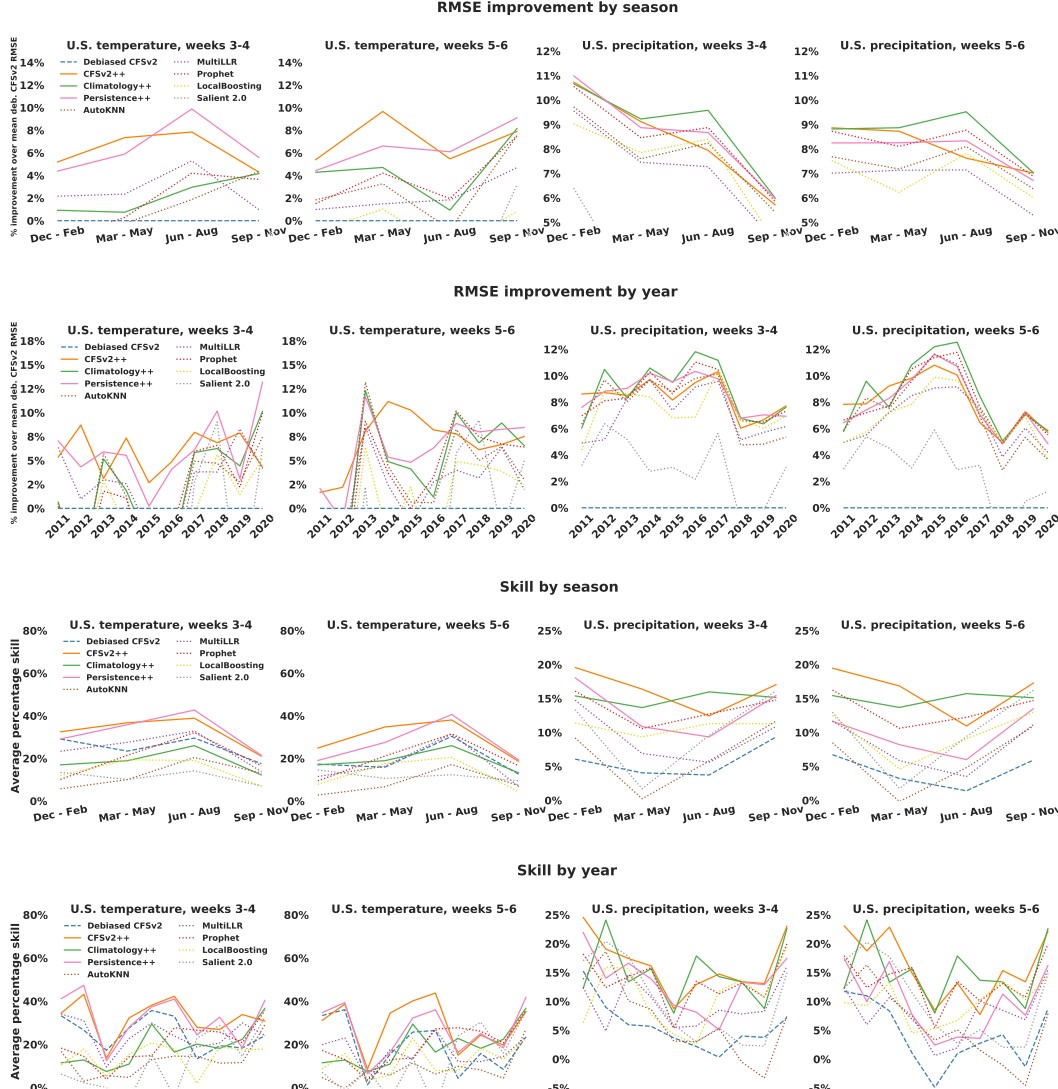

Figure 3: Per season and per year average skill and improvement over mean debiased CFSv2 RMSE across the contiguous U.S. and the years 2011–2020. Despite their simplicity, the ABC models (solid lines) consistently outperform debiased CFSv2 and the state-of-the-art learners (dotted lines).

**Western U.S. Competition Results**     Finally, we evaluate our models on the exact geographic region and target dates of the recent Subseasonal Climate Forecast Rodeo II competition. Specifically, we produce forecasts for the Western U.S. region, delimited by latitudes 25N to 50N and longitudes 125W to 93W, at a $1° \times 1°$ resolution for a total of $G = 514$ grid points. Forecasts were issued every two weeks for a yearlong period with initial issuance date October 29, 2019 and final issuance date October 27, 2020, leading to a noisier evaluation with only 26 observations.

Table 8 in Appendix C.7 compares the predictive accuracy of the models studied in this work with the accuracy of the contest baselines (debiased CFSv2, Climatology, and, for precipitation only, the Rodeo I Salient model of [59]) and the performance of the top competitors for each task. For temperature weeks 3-4, Persistence++ provides a 16.59% improvement over the mean debiased CFSv2 RMSE, outperforming the contest baselines, the state-of-the-art learning methods, and all but two of the competitors (the top three competitors improved by 17.12%, 16.67%, and 15.47%). For temperature weeks 5-6, CFSv2++ provides a 9.26% improvement over debiased CFSv2 and, despite its simplicity, outperforms the contest baselines, the state-of-the-art learning methods, and all of the

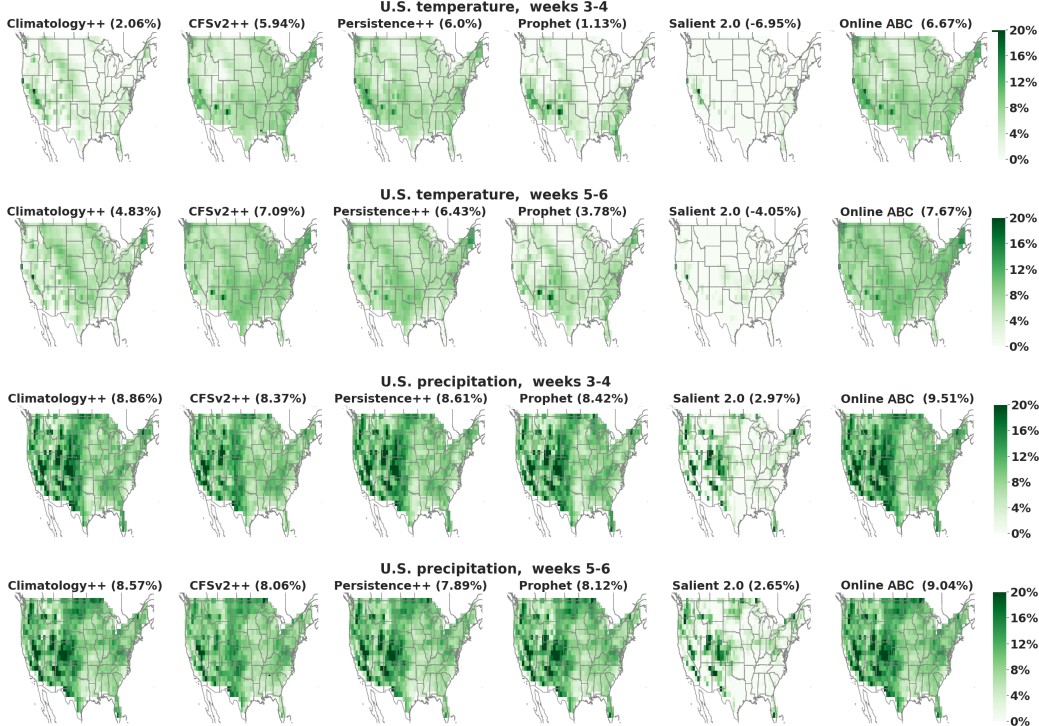

Figure 4: Percentage improvement over mean debiased CFSv2 RMSE in the contiguous U.S. over 2011–2020. White grid points indicate negative or 0% improvement.

competitors in the subseasonal forecasting competition (the top competitor improved by 8.47%). On this task, the Uniform and Online ABC ensembles also outperform all competitors.

On the precipitation tasks, the Salient baseline performed strongly and ultimately placed second and fourth respectively for the weeks 3-4 and weeks 5-6 tasks. Our Salient 2.0 model also performs remarkably well, outscoring all contestants and baselines with 12.65% improvement for weeks 3-4. For comparison, the top competitors for weeks 3-4 and weeks 5-6 improved by 11.54% and 8.63% respectively. Our Uniform ABC ensemble outperforms the remaining baselines and state-of-the-art learning methods but falls short of the exceptional Salient performance. In this setting, applying the adaptive online learning ensemble to the union of the ABC models and the state-of-the-art learners (denoted by Online ABC + Learning in Table 8) allows the user to exploit the irregular complementary benefits of the learning methods yielding 12.52% and 8.18% improvements in weeks 3-4 and 5-6.

## 6 Conclusion

In this work, we release SubseasonalClimateUSA, a dataset for subseasonal forecasting in the U.S. It is routinely updated and can be accessed as a Python package. The dataset includes a variety of features that are relevant at the subseasonal timescale, including precipitation, temperature, surface pressure, relative humidity, geopotential height, sea surface temperature, sea ice concentration, MJO, and MEI. We use this dataset to train and benchmark multifarious models, including deep learning solutions, dynamical models, and simple learned corrections, as well ensembling strategies. Our experiments with temperature and precipitation forecasting in the contiguous U.S. show that simple learning-based corrections to operational dynamical models yield low-cost strategies that are 10% more accurate and 329% more skillful than the U.S. operational CFSv2 and outperform state-of-the-art machine and deep learning methods, as well as the leading ECMWF dynamical model. Overall, we find that the SubseasonalClimateUSA dataset facilitates both the training and benchmarking of subseasonal forecasting models and hope that it will stimulate new advances in extended range forecasting.

## Acknowledgments and Disclosure of Funding

We acknowledge the agencies that support the SubX system, and we thank the climate modeling groups (Environment Canada, NASA, NOAA/NCEP, NRL and University of Miami) for producing and making available their model output. NOAA/MAPP, ONR, NASA, NOAA/NWS jointly provided coordinating support and led development of the SubX system. This work is based on S2S data. S2S is a joint initiative of the World Weather Research Programme (WWRP) and the World Climate Research Programme (WCRP). The original S2S database is hosted at ECMWF as an extension of the TIGGE database. This work was supported by Microsoft AI for Earth (S.M. and G.F.); the Climate Change AI Innovation Grants program (S.M., P.O., G.F., J.C., E.F., and L.M.), hosted by Climate Change AI with the support of the Quadrature Climate Foundation, Schmidt Futures, and the Canada Hub of Future Earth; FAPERJ (Fundação Carlos Chagas Filho de Amparo à Pesquisa do Estado do Rio de Janeiro) grant SEI-260003/001545/2022 (P.O.); NOAA grant OAR-WPO-2021-2006592 (G.F., J.C., and L.M.); and the National Science Foundation grant PLR-1901352 (J.C.).

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

# Supplementary Material for **SubseasonalClimateUSA: A Dataset for Subseasonal Forecasting and Benchmarking**

## A  SubseasonalClimateUSA **Supplementary Details**

The `subseasonal_data` Python package provides a detailed description of the SubseasonalClimateUSA dataset contents, sources, and processing steps at the following URL:

[https://github.com/microsoft/subseasonal_data/blob/main/DATA.md](https://github.com/microsoft/subseasonal_data/blob/main/DATA.md)

### A.1  Computational Environment Details

The SubseasonalClimateUSA update pipeline runs in a Docker container on an `Azure E16-4ds_v4` instance with 4 vCPUs/cores and 128 GB of RAM. The update pipeline runs in under 6 hours at a total cost of $5. All benchmarking experiments were run on the Massachusetts Institute of Technology engaging cluster ([https://engaging-web.mit.edu/eofe-wiki/](https://engaging-web.mit.edu/eofe-wiki/)).

### A.2  Western U.S. Competition Data Details

Following [23], for the Western U.S. competition experiments of Section 5, the sea surface temperature and sea ice concentration variables were formed by identifying the top three principal components for each variable restricted to the Pacific basin region (20S to 65N, 150E to 90W) using loadings from 1981-2010.

## B  Model Implementation Details

This section describes the implementation details for each learning model, including the training, hyperparameter tuning, and validation protocols. All models were implemented in Python 3.

### B.1  Climatology++

Climatology++ was trained and tuned following the protocol described in [39]; see Algorithm 1. Figure 5 displays the selected window length (the span $s$) and number of years $Y$ for each target date in 2011–2020 when forecasting for the contiguous U.S. We see that the temperature models preferred fewer training years and larger windows around the target day in recent history but focused more exclusively on the target day of year (via a span of 0) in 2013-2016 and preferred more training years in 2011. Meanwhile, the precipitation models selected the largest available window (corresponding to higher bias but lower variance estimates) for nearly every target date.

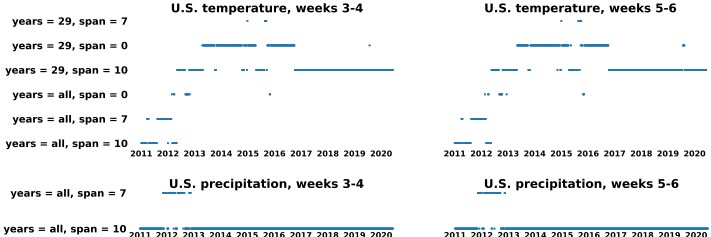

Figure 5: Climatology++ hyperparameters automatically selected for each target date in 2011–2020.

### B.2  CFSv2++

CFSv2++ was trained and tuned following the protocol described in [39]; see Algorithm 2. Figure 6 displays the selected window length (the span $s$), lead time range, and issuance date count for each target date in 2011–2020 when forecasting for the contiguous U.S. In each task, we observe a

**Algorithm 1** Climatology++

**input** test date $t^\star$; # train years $Y$; span $s$; loss $\in \{\text{RMSE,MSE}\}$; training set ground truth $(\mathbf{y}_t)_{t \in \mathcal{T}}$
**initialize** days per year $D = 365.242199$
$\mathcal{S} = \{t \in \mathcal{T} : \texttt{year\_diff} \coloneqq \lfloor \frac{t^\star - t}{D} \rfloor \leq Y \text{ and } \texttt{day\_diff} \coloneqq \frac{365}{2} - |\lfloor (t^\star - t) \mod D \rfloor - \frac{365}{2}| \leq s\}$
**output** $\text{argmin}_\mathbf{y} \sum_{t \in \mathcal{S}} \text{loss}(\mathbf{y}, \mathbf{y}_t)$

significant amount of variability in the optimal span, lead, and date count selections, highlighting the value of adaptive ensembling and debiasing over the static ensembling and debiasing strategies employed by standard debiased CFSv2.

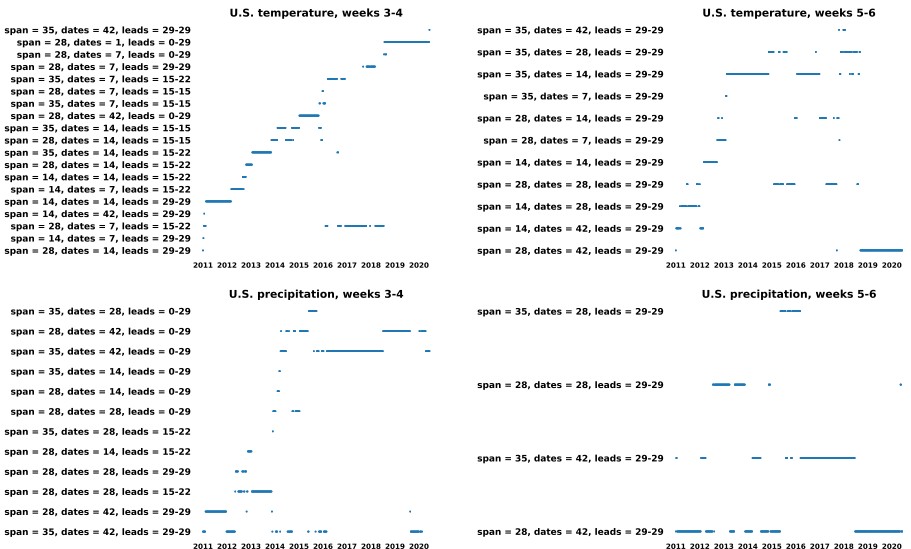

Figure 6: CFSv2++ hyperparameters automatically selected for each target date in 2011–2020.

**Algorithm 2** CFSv2++

**input** test date $t^\star$; lead time $l^\star$; # issuance dates $d^\star$; span $s$; training set ground truth and CFSv2 forecasts $(\mathbf{y}_t, \mathbf{f}_{t,l})_{t \in \mathcal{T}, l \in \mathcal{L}}$
**initialize** days per year $D = 365.242199$; # training years $Y = 12$
$\mathcal{S} = \{t \in \mathcal{T} : \texttt{year\_diff} \coloneqq \lfloor \frac{t^\star - t}{D} \rfloor \leq Y \text{ and } \texttt{day\_diff} \coloneqq \frac{365}{2} - |\lfloor (t^\star - t) \mod D \rfloor - \frac{365}{2}| \leq s\}$
// Form CFSv2 ensemble forecast across issuance dates and lead times $l \in \mathcal{L}$
**for** training and test dates $t \in \mathcal{S} \cup \{t^\star\}$ **do**
$\bar{\mathbf{f}}_t = \text{mean}((\mathbf{f}_{t-l^\star-d+1,l})_{1 \leq d \leq d^\star, l \in \mathcal{L}})$
**output** $\bar{\mathbf{f}}_{t^\star} + \text{mean}((\mathbf{y}_t - \bar{\mathbf{f}}_t)_{t \in \mathcal{S}})$

### B.3 Persistence++

Persistence++ was trained and tuned following the protocol described in [39]; see Algorithm 3. Figures 7 to 10 display the learned Persistence++ regression weights for the final target date in 2020 for each of the four contiguous U.S. forecasting tasks. In each case, we observe significant spatial variation in the optimal weights used to combine lagged measurements, climatology, and CFSv2 ensemble forecasts.

---

**Algorithm 3** Persistence++

---

**input** lead time $l^\star$; training set ground truth, climatology, and dynamical forecasts $(\mathbf{y}_t, \mathbf{c}_t, \mathbf{f}_{t,l})_{t\in\mathcal{T}, l\in\mathcal{L}}$

**initialize** forecast period length $L = 14$

    // Form dynamical ensemble forecast across subseasonal lead times $l \geq l^\star$

    **for** training dates $t \in \mathcal{T}$ **do**

        $\bar{\mathbf{f}}_t = \text{mean}((\mathbf{f}_{t,l})_{l \geq l^\star})$

    // Combine ensemble forecast, climatology, and lagged measurements

    **for** grid points $g = 1$ **to** $G$ **do**

        $\hat{\boldsymbol{\beta}}_g \in \text{argmin}_{\boldsymbol{\beta}} \sum_{t\in\mathcal{T}} (y_{t,g} - \boldsymbol{\beta}^\top[1, c_{t,g}, y_{t-l^\star-L-1,g}, y_{t-2l^\star-L-1,g}, \bar{f}_{t-l^\star-1,g}])^2$

**output** coefficients $(\hat{\boldsymbol{\beta}}_g)_{g=1}^G$

---

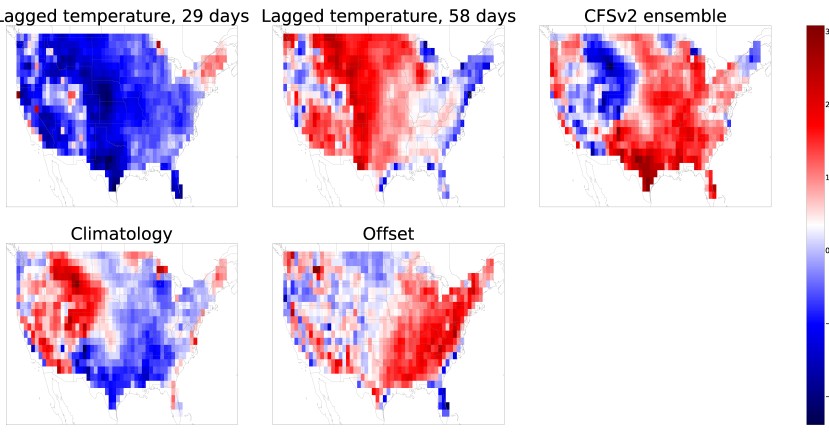

Figure 7: Spatial variation in Persistence++ learned regression weights when forecasting temperature in weeks 3-4 for the final target date, December 23, 2020.

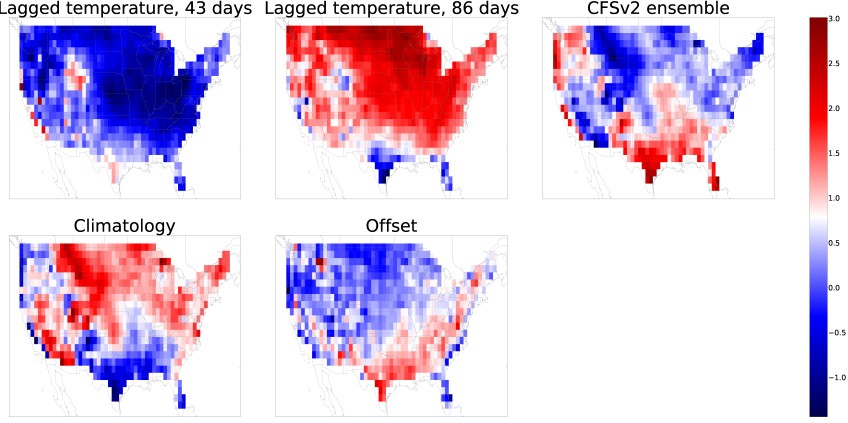

Figure 8: Spatial variation in Persistence++ learned regression weights when forecasting temperature in weeks 5-6 for the final target date, December 23, 2020.

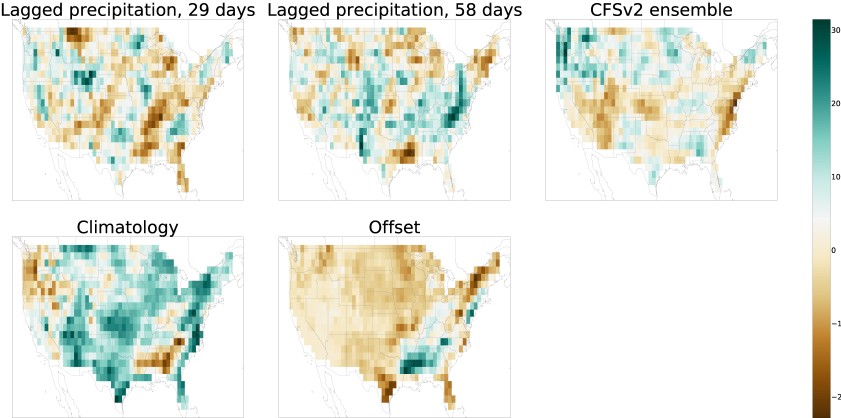

Figure 9: Spatial variation in Persistence++ learned regression weights when forecasting precipitation in weeks 3-4 for the final target date, December 23, 2020.

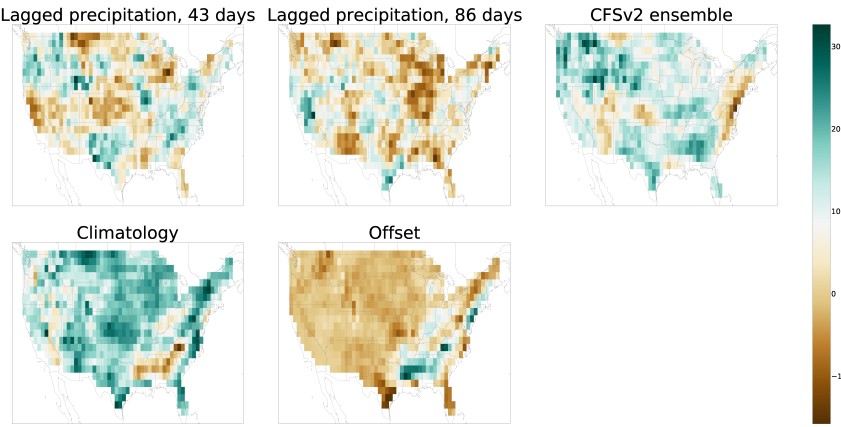

Figure 10: Spatial variation in Persistence++ learned regression weights when forecasting precipitation in weeks 5-6 for the final target date, December 23, 2020.

### B.4 AutoKNN

The AutoKNN model of [23] was part of a winning solution in the Subseasonal Climate Forecast Rodeo I [46] and was shown to outperform deep fully connected neural networks [19]. AutoKNN first identifies a set of historical dates most similar to the target date and then forecasts a weighted locally linear combination of the anomalies measured on similar dates and recent dates. Our implementation matches that of [23] but adapts the model to target our primary RMSE objective by (i) using mean (negative) RMSE as the similarity measure instead of mean skill, (ii) using raw measurement vectors $\mathbf{y}_t$ instead of anomaly vectors $\mathbf{a}_t$, and (iii) using equal datapoint weights in the final local linear regression.

**Training** The k-nearest neighbors (KNN) step of AutoKNN identifies a set of historical dates most similar to the target date and while the autoregression step forecasts a weighted locally linear combination of the anomalies measured on similar dates and recent dates. For a given target date $t^\star$ and lead time $l^\star$, the AutoKNN training set is restricted to data fully observable one day prior to

the issuance date, that is, to dates $t \leq t^\star - l^\star - L - 1$ where $L = 14$ represents the forecast period length.

**Tuning**    All hyperparameters were set to the default values specified in [23].

### B.5    Informer

The Informer [79], a transformer-based deep learning model, was retrained every four months to predict temperature from past temperature and precipitation from past precipitation independently at each grid point.

**Features**    For a given grid point and target date $t^\star$, the input features used to construct a forecast are the lagged target variable observations from dates $t_{\text{last}}, t_{\text{last}} - 1, t_{\text{last}} - 2, \cdots, t_{\text{last}} - 95$ for where $t_{\text{last}} = t^\star - l^\star - L$ represents the last complete observation prior to $t^\star$ and $L = 14$ represents the forecast period length.

**Training**    We divide the set of target dates in 2011–2020 into consecutive, non-overlapping four-month blocks and retrain the Informer model after every four-month block. For a given lead time $l^\star$, grid point, and four-month block beginning with date $t^\star$:

1. The training set is chosen to start at most $10,000$ days before $t$ (or at the beginning of the training set, whichever is later) and then ends $301$ days before $t$.

2. The validation set is chosen to start $300$ days before $t$ and to end on the date prior to $t^\star - l^\star - L$.

3. We use early stopping with patience equal to three to determine when to stop the training: when we have three consecutive epochs $e + 1, e + 2, e + 3$ with validation loss no lower than that of epoch $e$, we terminate training and use the model at epoch $e$ as the final trained model.

4. We use the trained model to generate forecasts for each target date in the four-month block.

**Tuning**    We use the default Informer architecture and hyperparameters for univariate time series forecasting: the model has 3 encoder layers, 2 decoder layers, and has an 8-headed attention with 7-dimensional keys and feed-forward layers with 1024 hidden units, and has GeLU activations [20].

### B.6    LocalBoosting

In recent subseasonal experiments of [19], boosted decision tree models yielded the best performance. Our boosted decision tree model, based on CatBoost [51], uses as features the value of 10 SubseasonalClimateUSA variables in a geographic region around the target grid point. This gives the algorithm enough flexibility to adapt the weights of the features to each particular grid point while still taking into account neighboring spatial information. The geographic region is determined by a bounding box of 2 degrees in each direction, and the 10 variables are chosen for each task via their predictive power on validation years.

**Training**    For a given target date $t^\star$ and lead time $l^\star$, the LocalBoosting training set $\mathcal{T}$ is restricted to data fully observable one day prior to the issuance date, that is, to dates $t \leq t^\star - l^\star - L - 1$ where $L = 14$ represents the forecast period length. LocalBoosting uses CatBoost [51] to regress, for each gridpoint and each date, the value of a set of lagged weather variables in a geographic region around the gridpoint.

**Tuning**    There are two hyperparameters to consider: (i) which lagged weather variables to use; and (ii) the number of neighborhood cells around a gridpoint to define the geographic region. Bounding boxes of with side length of 2 or 3 cells were considered. Larger sizes were computationally infeasible. In each case, the 10 or 20 most important features in the SubseasonalClimateUSA dataset were considered. Here, features were chosen by their performance over 2001-2010 in terms of RMSE.

For each target date, LocalBoosting is run with the hyperparameter configuration that achieved the smallest mean RMSE over the preceding 3 years. See Figure 11 for a visualization of the hyperparameters automatically selected for each target date in 2011–2020.

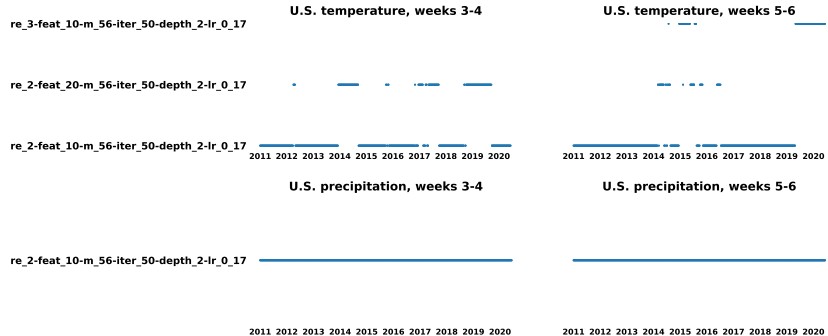

Figure 11: LocalBoosting hyperparameters automatically selected for each target date in 2011–2020.

## B.7 MultiLLR

The MultiLLR model of [23] was also part of a winning solution in the Subseasonal Climate Forecast Rodeo I [46] and has since been used to improve subseasonal precipitation prediction in China [67]. For each target date, MultiLLR uses a backward stepwise procedure to select the most predictive features and then applies local linear regression for each grid point to combine those features into a final prediction. Our implementation matches that of [23] but adapts the model to target our primary RMSE objective by using mean (negative) RMSE instead of mean skill as the feature selection criterion. In addition, we replace their MEI features with corresponding MEI.v2 features and their monthly dynamical forecast features with daily debiased CFSv2 forecasts.

**Training** For a given target date $t^\star$ and lead time $l^\star$, the MultiLLR training set is restricted to data fully observable one day prior to the issuance date, that is, to dates $t \leq t^\star - l^\star - L - 1$ where $L = 14$ represents the forecast period length. The coarse-grained dynamical input feature `nmme_wo_ccsm3_nasa` of [23] was replaced with the daily debiased CFSv2 forecast features

- `subx_cfsv2_tmp2m-14.5d_shift15` and `subx_cfsv2_tmp2m-0.5d_shift15` for predicting temperature at weeks 3-4,
- `subx_cfsv2_tmp2m-28.5d_shift29` and `subx_cfsv2_tmp2m-0.5d_shift29` for predicting temperature at weeks 5-6,
- `subx_cfsv2_precip-14.5d_shift15` and `subx_cfsv2_precip-0.5d_shift15` for predicting precipitation at weeks 3-4, and
- `subx_cfsv2_precip-28.5d_shift29` and `subx_cfsv2_precip-0.5d_shift29` for predicting precipitation at weeks 5-6.

**Tuning** All hyperparameters were set to the default values specified in [23], save for the tolerance parameter which was set to $0.001$ to accommodate the new RMSE selection criterion.

## B.8 N-BEATS

N-BEATS [48], a neural network for time series data, is retrained N-BEATS every two months to predict temperature from past temperature and precipitation from past precipitation independently at each grid point.

**Features** For a given grid point and target date $t^\star$, the input features used to construct a forecast are the lagged target variable observations from dates $t_{last}, t_{last} - 4, t_{last} - 8, \cdots, t_{last} - 48$ for where $t_{last} = t^\star - l^\star - L$ represents the last complete observation prior to $t^\star$ and $L = 14$ represents the forecast period length.

**Training** We divide the set of target dates in 2011–2020 into consecutive, non-overlapping two-month blocks and retrain the N-BEATS model after every two-month block. For a given lead time $l^\star$, grid point, and two-month block beginning with date $t^\star$:

1. We train on all dates $t \leq t^\star - l^\star - L$.

2. For the initial two-month block, we train for 30 epochs.

3. For subsequent two-month blocks, we initialize our model weights to the learned weights from the prior block and then fine-tune for 8 epochs.

4. We use the trained model to generate forecasts for each target date in the two-month block.

**Tuning**   We use the default N-BEATS architecture and hyperparameters for univariate time series forecasting [48]. The N-BEATS model has two stacks, where each stack is used to understand different patterns. Each stack consists of three blocks, which themselves are each comprised of six fully connected layers with ReLU activations. We used the Adam optimizer with learning rate $0.001$ and a batch size of $512$.

## B.9   Prophet

The Prophet model of [62] was one of the winning solutions in the Subseasonal Forecast Rodeo II [46]. Prophet is an additive regression model for time-series forecasting that predicts weekly and yearly seasonal trends on top of a piecewise linear or logistic growth curve. We trained the model to predict each grid point independently with yearly seasonality enabled (to capture predictable whether trends) and weekly seasonality disabled.

**Training**   Prophet takes as input a sequence of (univariate) time-series values and then predicts the next $k$ dates from those, arbitrarily far in the future.

First, we split the problem into a many univariate time-series prediction problems. When evaluating, we consider periods of 4 moths (e.g. January 2010 - April 2010), train the model on all available historical temperatures (which may depend on the lead time) at that grid point, and make predictions for that four month period. We then run this for all relevant periods of four months.

**Tuning**   The prophet model is trained in the univariate mode with yearly seasonality on (to capture predictable weather trends), weekly seasonality off (as weekends are unlikely to be special).

## B.10   Salient 2.0

We developed the Salient 2.0 model based on Salient [59], a winning solution for the Subseasonal Forecast Rodeo I [46]. Salient consists of an ensemble of feed-forward fully-connected neural networks, using historical sea surface temperature (SST) data from 1990 to 2017 and an encoding of the day of the year as features. Salient's training protocol follows a multi-task learning framework [56]. It starts by training 50 randomly generated fully connected neural networks, each of which provides a prediction for the average temperature and accumulated precipitation at 3, 4, 5, and 6 weeks ahead, at every grid cell. The forecasts are then obtained by combining the predictions for weeks 3 and 4 and for weeks 5 and 6. The final ensemble model forecasts correspond to the mean of the top 10 ensemble members with the lowest validation error.

For Salient 2.0 in this work, the input features were augmented with geopotential heights at different pressure levels (10, 100, 500 and 850 hPa) along with MEI and MJO indices. In addition, instead of training the ensemble on the whole of the 1990-2017 data, a sequence of models was trained using data up until each of the years in our validation period of 2010-2020. These submodels with earlier training data cut-offs were then used to generate hindcasts that informed the model's tuning decisions.

Salient 2.0 relies on two sources of sea surface temperature training data. The first data source is the weekly sea surface temperature from NOAA [54] and covers dates from 17 January 1990 to 02 February 2017. This dataset contains weekly data cenetered around Wednesdays and has a $1° \times 1°$ resolution. It was re-gridded to a $4° \times 4°$ using spline interpolation of order equal to 2, under the Python Scipy package [63].

For dates from February 2017 to present, a second source of sea surface temperature data from the MET Office [30] is used. This dataset has a daily temporal resolution and is averaged to obtain weekly data centered around Wednesdays. This data is initially downloaded in a $0.25° \times 0.25°$ spatial resolution and is re-gridded to a $4° \times 4°$. A linear interpolation is then used to fill in any missing values (under the Python Scipy package [63]).

**Training**  Salient 2.0 is an ensemble of 50 feed-forward fully connected neural networks. All 50 neural networks are trained on a pre-determined combination of input features, including weekly sea surface temperature, MEI, as well as the phase and amplitude features of MJO. The combinations of input features considered are:

- `d2wk_cop_sst`: sea surface temperature,
- `d2wk_cop_sst_mei`: sea surface temperature and ENSO,
- `d2wk_cop_sst_mjo`: sea surface temperature and MJO,
- `d2wk_cop_sst_mei_mjo`: sea surface temperature, ENSO and MJO.

In total, four ensemble models, corresponding to the four input feature combinations above, each including 50 neural networks, are trained. Each ensemble model is trained in a rolling fashion, where the start year of the training dataset is 1990 and the ensemble is trained up until each year in the range [2006, 2019], where a model ending on a year $y$ is used to generate forecasts for target dates with year $y + 1$.

For each of the 50 neural networks within a given ensemble model, the input features can further be augmented using a time vector obtained by converting dates to a float representing the fraction of the year passed by that date. The addition of the time vector is decided by generating a random integer in the range $[0, 1]$, with 1 corresponding to the addition of the time vector and 0 otherwise. Additionally, for each of the 50 neural networks, the input feature vector for an individual training example consists of a concatenation of the prior 10 weeks of data.

For each of the 50 neural networks within a given ensemble model, the output consists of a prediction for the average temperature and accumulated precipitation at 3, 4, 5 and 6 weeks ahead. The predictions for weeks 3 and 4 and the predictions for weeks 5 and 6 are combined separately, by averaging temperatures and summing precipitation. The top 10 neural networks with the lowest validation error are selected as the final ensemble members. The final predictions for each ensemble model correspond to the mean of the 10 selected ensemble members.

**Tuning**  Each of the 50 neural networks within a given ensemble model is trained using a batch size equal to 128 and a train ratio equal to 0.89. In addition, each neural network's hyperparamaters are randomly generated, with the number of epochs sampled in the range $[100, 500]$, the number of layers in the range $[3, 7]$, and the number of units per layer in the range $[100, 600]$.

At test time, for each target date, Salient 2.0 is run using the ensemble model that achieved the smallest mean RMSE over the preceding 3 years. Figure 12 shows which ensemble models were used to generate predictions for which target dates.

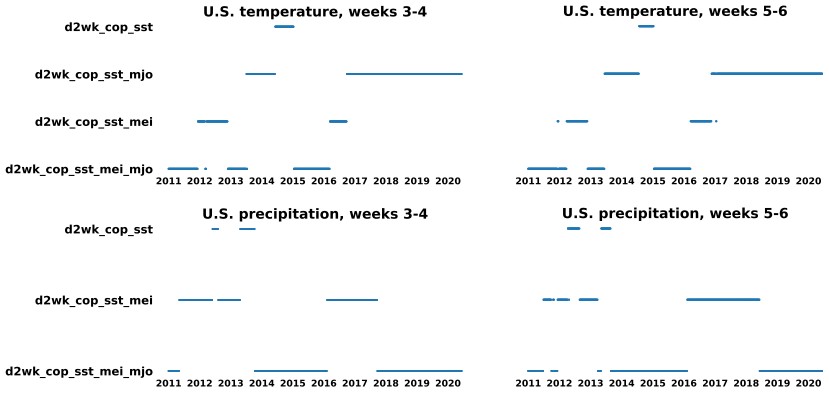

Figure 12: Salient 2.0 hyperparameters automatically selected for each target date in 2011–2020.

## B.11  Uniform Ensemble

We consider two Uniform Ensemble models, where the ensemble is produced using a set of models $C$ as input:

1. **Uniform ABC**: $C = \{$ Climatology++, CFSv2++, Persistence++ $\}$
2. **Uniform ABC + Learning**: $C = \{$ Climatology++, CFSv2++, Persistence++, LocalBoosting, MultiLLR, AutoKNN, Prophet, Salient 2.0 $\}$, the ABC models plus all learning models save the very low performing Informer and N-BEATS models.

Uniform ensemble forecasts are produced as the uniform or unweighted average of input models. Letting $X_{t,c}$ be the forecast made by model $c \in C$ on target date $t$, we produce the forecast: $\hat{\mathbf{y}}_t = \frac{1}{|C|} \sum_{c \in C} X_{t,c}$.

### B.12 Online Ensemble

We consider two Online Ensemble models, where the ensemble is produced using a set of models $C$ as input:

1. **Online ABC**: $C = \{$Climatology++, CFSv2++, Persistence++$\}$
2. **Online ABC + Learning**: $C = \{$ Climatology++, CFSv2++, Persistence++, LocalBoosting, MultiLLR, AutoKNN, Prophet, Salient 2.0 $\}$, the ABC models plus all learning models save the very low performing Informer and N-BEATS models.

To learn a time-dependent adaptive ensemble weight $\mathbf{w}_t$, we employ the online learning method presented in [15]. We applied the AdaHedgeD algorithm with the recommended `recent_g` optimism setting. The algorithm was run with a delay parameter of $D = 2$ for the 3-4 weeks horizon tasks and $D = 3$ for the 5-6 weeks horizon tasks. We ran the online learning algorithm over the full set of target dates $T = 520$, without performing the yearly resetting suggested in the original implementation. The learner optimized the RMSE loss over gridpoints in the region of interest, as described in the experimental details of [15].

Online ensemble forecasts are produced as the weighted average of input models, with weight $\mathbf{w}_t$ determined by the online learning algorithm. Letting $X_{t,c}$ be the forecast made by model $c \in C$ on target date $t$ and $\mathbf{w}_t$ be the weights produced by AdaHedgeD, we produce the forecast: $\hat{\mathbf{y}}_t = \sum_{c \in C} \mathbf{w}_{t,c} * X_{t,c}$.

### B.13 Debiased ECMWF

We implement the operational ECMWF bias correction protocol detailed in [69]. For each target forecast date, we debias both our ECMWF control and ensemble forecasts using the last 20 years of reforecasts with dates within $\pm 6$ days from the target forecast date. The average of the 1 control and 10 ensemble reforecasts on the 1.5x1.5 degree grid are used for debiasing.

## C   Supplementary Results

### C.1   Percentage Improvement over Meteorological Baselines

To highlight the improvement of individual ABC models over their traditional counterparts, Figures 13 and 14 show the percentage RMSE improvements of Climatology++, CFSv2++, and Persistence++ relative to their respective baselines Climatology, debiased CFSv2, and Persistence by season and by year.

For all four tasks, the ABC models are consistently better across seasons and years. The result is particularly striking for Persistence++ and highlights the value of integrating lagged measurements, numerical weather prediction, and climatology. Figures 13 and 14 show the per season and per year improvement of each ABC model over its corresponding baseline across the contiguous U.S. and the years 2011–2020. Note the learned ABC benchmarks yield consistent improvements in mean RMSE.

### C.2   Yearly Percentage Improvement over Mean Debiased CFSv2 RMSE

Tables 3 and 4 present the yearly improvement of each model over debiased CFSv2, as measured by mean RMSE across the contiguous U.S. in the years 2011–2020.

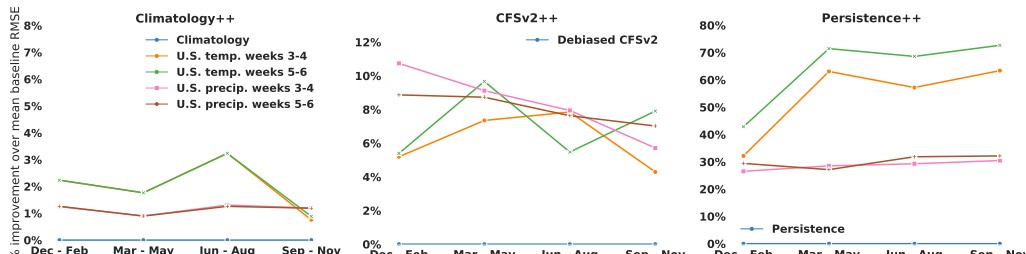

Figure 13: Per season improvement of each ABC model over its corresponding baseline across the contiguous U.S. and the years 2011–2020. The learned ABC benchmarks yield consistent improvements in mean RMSE.

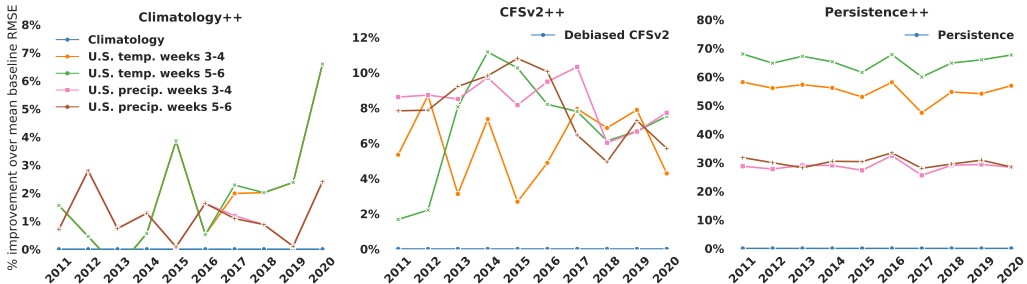

Figure 14: Per year improvement of each ABC model over its corresponding baseline across the contiguous U.S. and the years 2011–2020. The learned ABC benchmarks yield consistent improvements in mean RMSE.

Table 3: Percentage improvement over mean debiased CFSv2 RMSE when forecasting temperature in the contiguous U.S. The best performing models within each class of models are shown in bold, while the best performing models overall are shown in green.

| Group | Model | 2011 | 2012 | 2013 | 2014 | 2015 | 2016 | 2017 | 2018 | 2019 | 2020 | Overall |
|---|---|---|---|---|---|---|---|---|---|---|---|---|
| | | | | | | Temperature, weeks 3–4 | | | | | | |
| Baselines | Climatology | −0.89±3.85 | −7.52±4.56 | 5.8±3.62 | 1.32±3.86 | −7.91±4.85 | −5.54±5.79 | 3.93±3.72 | 4.32±3.62 | 2.1±4.43 | 3.81±4.36 | 0.13±1.35 |
| | Persistence | −122.09±14.57 | −117.74±11.91 | −120.23±16.63 | −115.49±19.12 | −112.3±18.61 | −128.89±17.9 | −78.66±12.11 | −98.39±13.72 | −111.15±23.56 | −101.39±16.54 | −109.94±5.29 |
| ABC | Climatology++ | 0.68±3.58 | −7.03±4.31 | 5.17±3.72 | 1.87±3.86 | −3.74±4.36 | −4.99±6.12 | 5.83±3.84 | 6.25±3.24 | 4.43±4.01 | 10.15±3.66 | 2.06±1.3 |
| | CFSv2++ | 5.35±3.63 | 8.71±3.04 | 3.12±3.26 | 7.36±3.85 | 2.68±3.38 | 6.17±3.86 | 7.94±3.28 | 6.87±2.77 | 7.88±2.89 | 4.29±3.29 | 5.92±1.05 |
| | Persistence++ | 7.06±3.21 | 4.33±3.37 | 5.91±3.04 | 5.54±3.77 | 0.23±3.67 | 4.11±4.19 | 6.1±3.31 | 10.16±2.54 | 3.12±3.2 | 13.21±2.79 | 6.0±1.06 |
| Learning | AutoKNN | 0.52±4.05 | −7.99±4.65 | 5.74±3.17 | 1.25±3.86 | −7.27±4.92 | −4.6±5.47 | 4.93±3.48 | 4.71±3.59 | 2.75±3.88 | 7.44±4.14 | 0.93±1.34 |
| | LocalBoosting | −3.74±4.23 | −6.89±4.57 | 2.92±2.98 | −1.92±4.19 | −5.86±4.28 | −5.23±5.7 | −0.6±3.76 | 5.62±3.15 | 1.45±3.18 | 5.19±3.54 | −0.76±1.23 |
| | MultiLLR | 6.37±3.53 | 0.97±3.85 | 3.0±3.47 | 2.57±3.54 | −2.43±3.03 | −7.23±5.75 | 3.83±3.68 | 3.8±3.25 | 8.3±3.08 | 4.15±3.09 | 2.45±1.17 |
| | Prophet | 0.19±4.22 | −7.89±4.41 | 1.81±3.83 | 1.07±4.4 | −7.88±4.82 | −2.5±4.9 | 5.95±4.07 | 6.59±3.52 | 2.22±4.69 | 9.97±4.18 | 1.13±1.36 |
| | Salient 2.0 | −7.6±4.9 | −13.08±5.82 | −7.81±4.41 | −22.21±6.47 | −11.02±4.4 | −19.04±6.79 | 2.71±3.44 | 9.03±3.05 | −10.54±6.2 | 6.48±5.16 | −6.95±1.76 |
| Ensembles | Uniform ABC | 6.23±3.41 | 4.08±3.34 | 6.32±2.91 | 6.57±3.69 | 1.12±3.73 | 4.39±4.17 | 7.34±3.25 | 9.14±2.78 | 7.08±3.06 | 11.79±2.79 | 6.46±1.05 |
| | Online ABC | 5.57±3.26 | 6.49±3.25 | 5.2±3.07 | 7.3±3.78 | 2.39±3.33 | 6.49±3.9 | 7.86±3.16 | 9.01±2.62 | 6.77±2.59 | 10.44±2.66 | 6.71±1.01 |
| | | | | | | Temperature, weeks 5–6 | | | | | | |
| Baselines | Climatology | −3.49±3.33 | −8.66±5.54 | 12.91±3.38 | 4.36±3.68 | 0.26±3.81 | 0.66±4.39 | 7.89±2.84 | 4.93±2.63 | 6.71±3.88 | −0.05±4.37 | 2.93±1.23 |
| | Persistence | −206.03±22.74 | −187.94±22.83 | −169.31±24.87 | −172.6±28.7 | −147.71±27.59 | −190.83±16.81 | −127.66±18.96 | −161.7±19.97 | −169.53±27.67 | −183.02±26.03 | −170.1±7.56 |
| ABC | Climatology++ | −1.88±3.47 | −8.17±5.4 | 12.32±3.64 | 4.89±3.3 | 4.11±3.39 | 1.18±4.4 | 10.0±2.54 | 6.84±2.5 | 8.93±3.45 | 6.55±3.85 | 4.83±1.18 |
| | CFSv2++ | 1.68±3.13 | 2.2±4.34 | 8.06±3.06 | 11.17±3.36 | 9.72±2.99 | 7.57±2.41 | 6.44±2.96 | 6.13±2.71 | 6.68±3.19 | 7.52±3.9 | 7.09±1.02 |
| | Persistence++ | 2.07±2.66 | −1.33±4.6 | 11.68±2.86 | 5.35±3.09 | 4.81±2.84 | 6.31±3.44 | 8.87±2.39 | 7.99±1.94 | 8.26±2.96 | 8.45±3.25 | 6.43±0.99 |
| Learning | AutoKNN | −3.49±3.43 | −9.57±5.81 | 13.19±3.28 | 4.41±3.83 | 0.68±3.99 | 0.58±4.2 | 8.25±2.71 | 4.99±2.51 | 6.47±3.45 | 2.98±4.04 | 3.22±1.25 |
| | LocalBoosting | −9.97±5.0 | −11.73±5.92 | 6.41±4.17 | −3.2±4.09 | 2.21±3.31 | −6.55±4.16 | 4.91±3.01 | 4.51±2.57 | 3.89±2.89 | 2.46±3.46 | −0.29±1.24 |
| | MultiLLR | −3.27±4.39 | −4.22±5.07 | 8.45±3.41 | 2.44±5.18 | −1.58±3.55 | 2.76±3.46 | 4.0±2.79 | 3.18±2.7 | 6.35±3.66 | 1.98±4.3 | 2.21±1.24 |
| | Prophet | −2.7±3.69 | −9.59±5.74 | 9.09±3.88 | 3.72±3.98 | 0.02±3.66 | 3.29±3.94 | 10.18±3.38 | 6.71±3.77 | 6.4±4.22 | 4.98±4.09 | 3.78±1.26 |
| | Salient 2.0 | −9.34±5.44 | −17.35±7.14 | 2.23±4.74 | −21.18±6.46 | −2.22±3.73 | −14.66±5.25 | 5.78±3.57 | 9.22±3.34 | −2.46±5.35 | 4.98±4.09 | −4.05±1.73 |
| Ensembles | Uniform ABC | 2.35±2.95 | −0.06±4.57 | 11.92±2.83 | 8.38±3.08 | 7.57±2.91 | 7.67±3.03 | 9.19±2.42 | 7.96±2.28 | 8.88±3.19 | 8.84±3.43 | 7.45±0.99 |
| | Online ABC | 2.15±2.75 | 0.94±4.65 | 11.06±2.91 | 9.41±3.16 | 9.19±2.71 | 7.04±2.76 | 8.49±2.59 | 7.96±2.22 | 8.6±2.92 | 8.87±3.49 | 7.54±0.98 |

Table 4: Percentage improvement over mean debiased CFSv2 RMSE when forecasting precipitation in the contiguous U.S. The best performing models within each class of models are shown in bold, while the best performing models overall are shown in green.

| Group | Model | 2011 | 2012 | 2013 | 2014 | 2015 | 2016 | 2017 | 2018 | 2019 | 2020 | Overall |
|---|---|---|---|---|---|---|---|---|---|---|---|---|
| | | | | | Precipitation, weeks 3-4 | | | | | | | |
| Baselines | Climatology | 5.37±1.87 | 7.91±1.69 | 7.62±1.65 | 9.42±2.11 | 9.42±1.82 | 10.36±1.8 | 10.1±2.05 | 5.95±1.28 | 6.26±1.31 | 5.35±1.49 | 7.79±0.54 |
| | Persistence | −29.66±4.88 | −26.25±4.43 | −28.19±3.37 | −26.57±3.83 | −24.47±5.02 | −32.78±5.7 | −21.24±5.48 | −31.45±3.72 | −31.65±4.46 | −29.99±4.1 | −28.27±1.4 |
| ABC | Climatology++ | 6.03±1.77 | 10.48±1.58 | 8.31±1.6 | 10.58±2.29 | 9.5±1.93 | 11.82±1.91 | 11.17±1.92 | 6.78±1.29 | 6.36±1.36 | 7.63±1.43 | 8.86±0.54 |
| | CFSv2++ | 8.62±1.78 | 8.72±1.74 | 8.5±1.44 | 9.71±1.99 | 7.31±1.57 | 9.32±1.49 | 10.32±1.82 | 6.03±1.54 | 6.66±1.29 | 7.73±1.24 | 8.26±0.51 |
| | Persistence++ | 7.6±1.67 | 8.82±1.65 | 9.04±1.45 | 10.19±1.99 | 9.55±1.78 | 10.33±1.72 | 9.77±1.84 | 6.79±1.33 | 7.04±1.3 | 6.93±1.26 | 8.61±0.51 |
| Learning | AutoKNN | 6.33±1.99 | 9.68±1.93 | 8.03±1.67 | 9.66±2.08 | 8.78±2.13 | 9.8±1.95 | 10.14±1.89 | 4.76±1.49 | 4.83±1.67 | 5.36±1.6 | 7.73±0.58 |
| | LocalBoosting | 4.37±1.82 | 8.88±1.58 | 8.65±1.65 | 8.4±2.39 | 6.81±2.17 | 6.87±2.43 | 10.36±1.84 | 6.65±1.37 | 6.03±1.46 | 6.96±1.6 | 7.36±0.56 |
| | MultiLLR | 4.91±1.8 | 5.15±2.29 | 8.3±1.37 | 9.69±1.94 | 7.38±1.66 | 9.15±1.52 | 9.57±1.79 | 5.14±1.22 | 5.71±1.25 | 6.16±1.36 | 7.12±0.51 |
| | Prophet | 6.96±1.8 | 8.1±1.78 | 8.3±1.59 | 10.29±2.21 | 8.63±1.99 | 11.04±1.82 | 10.48±1.97 | 6.6±1.28 | 6.5±1.36 | 7.33±1.45 | 8.42±0.53 |
| | Salient 2.0 | 3.2±2.45 | 6.37±2.33 | 5.18±2.17 | 2.76±2.67 | 3.07±3.0 | 2.16±2.73 | 5.64±2.28 | −1.06±2.05 | −0.21±2.24 | 3.1±1.93 | 2.97±0.78 |
| Ensembles | Uniform ABC | 8.34±1.73 | 10.35±1.62 | 9.47±1.37 | 11.01±2.03 | 9.9±1.76 | 11.46±1.61 | 11.13±1.84 | 7.21±1.33 | 7.4±1.34 | 8.26±1.26 | 9.45±0.5 |
| | Online ABC | 8.82±1.81 | 10.24±1.64 | 9.35±1.46 | 10.94±2.07 | 10.01±1.68 | 11.79±1.68 | 11.18±1.89 | 7.16±1.3 | 7.3±1.34 | 8.28±1.24 | 9.5±0.5 |
| | | | | | Precipitation, weeks 5-6 | | | | | | | |
| Baselines | Climatology | 5.1±1.57 | textbf6.98±1.46 | textbf6.93±1.56 | textbf9.67±1.43 | textbf12.12±1.74 | textbf11.09±1.87 | textbf7.45±1.31 | textbf4.22±0.95 | textbf6.97±1.33 | textbf3.52±1.36 | textbf7.51±0.48 |
| | Persistence | −37.12±5.31 | −32.23±4.32 | −27.77±3.88 | −29.87±3.39 | −26.92±4.95 | −34.16±5.52 | −29.11±4.56 | −35.06±5.24 | −34.24±4.6 | −33.06±3.93 | −31.92±1.43 |
| ABC | Climatology++ | 5.78±1.67 | 9.59±1.44 | 7.61±1.52 | textbf10.82±1.53 | textbf12.2±1.91 | 12.55±1.95 | 8.46±1.3 | textbf5.06±0.92 | 7.06±1.41 | textbf5.84±1.39 | textbf8.57±0.49 |
| | CFSv2++ | 7.84±1.31 | 7.87±1.53 | 9.22±1.4 | 9.82±1.38 | 10.5±1.7 | 10.13±1.3 | 6.46±1.32 | 4.94±1.16 | textbf7.28±1.23 | 5.71±1.25 | 8.03±0.44 |
| | Persistence++ | 6.44±1.33 | 7.42±1.46 | 8.28±1.4 | 9.71±1.32 | 11.62±1.79 | 10.71±1.77 | 7.05±1.23 | 4.83±0.96 | 7.19±1.23 | 4.85±1.19 | 7.89±0.45 |
| Learning | AutoKNN | 5.85±1.59 | textbf8.31±1.76 | 7.5±1.44 | 9.57±1.48 | textbf11.64±2.05 | 10.87±1.9 | 6.94±1.33 | 2.85±1.09 | 5.34±1.4 | 3.66±1.38 | 7.33±0.5 |
| | LocalBoosting | 4.94±1.71 | 5.76±1.55 | 7.24±1.58 | 7.84±1.66 | 9.89±1.98 | 9.64±2.02 | 7.28±1.29 | textbf4.97±1.07 | 6.94±1.44 | 3.59±1.51 | 6.89±0.49 |
| | MultiLLR | 4.94±1.45 | 5.51±1.63 | 7.2±1.52 | 8.49±1.32 | 9.07±1.74 | 9.17±1.73 | 7.49±1.32 | 3.85±1.06 | 6.04±1.3 | 4.18±1.45 | 6.65±0.46 |
| | Prophet | textbf6.65±1.56 | 7.19±1.61 | 7.71±1.51 | textbf10.53±1.5 | 11.38±1.97 | textbf11.77±1.86 | textbf7.64±1.32 | 4.84±0.96 | textbf7.19±1.48 | textbf5.52±1.46 | textbf8.12±0.48 |
| | Salient 2.0 | 2.93±2.06 | 5.39±2.14 | 4.53±1.88 | 3.01±2.09 | 5.89±2.97 | 2.89±2.53 | 3.22±2.11 | −3.3±1.68 | 0.51±1.86 | 1.24±1.89 | 2.65±0.7 |
| Ensembles | Uniform ABC | 7.7±1.39 | textbf9.35±1.44 | textbf9.19±1.29 | 11.0±1.35 | 12.39±1.79 | 12.26±1.54 | 8.05±1.2 | 5.58±0.94 | 7.89±1.31 | 6.35±1.22 | 9.05±0.44 |
| | Online ABC | textbf7.79±1.39 | 9.35±1.46 | 9.06±1.38 | 11.06±1.36 | 12.17±1.71 | textbf12.31±1.59 | textbf8.23±1.27 | 5.49±0.93 | 7.72±1.32 | 6.46±1.22 | 9.03±0.43 |

## C.3 Yearly Average Skill

Tables 5 and 6 present the yearly average skill of each model across the contiguous U.S. in the years 2011–2020.

Table 5: Average percentage skill when forecasting temperature in the contiguous U.S. The best performing models within each group are shown in bold, while the best performing models overall are shown in green.

| Group | Model | 2011 | 2012 | 2013 | 2014 | 2015 | 2016 | 2017 | 2018 | 2019 | 2020 | Overall |
|---|---|---|---|---|---|---|---|---|---|---|---|---|
| | | | | | Temperature, weeks 3-4 | | | | | | | |
| Baselines | Deb. CFSv2 | **33.26±5.13** | **26.66±6.65** | **17.32±5.58** | **27.66±5.85** | **35.73±5.71** | **33.07±5.77** | **13.36±5.92** | **19.42±6.07** | **18.39±6.12** | **24.14±6.43** | **24.94±1.88** |
| | Persistence | 27.76±5.36 | -1.64±7.59 | -2.16±7.09 | 17.98±7.15 | 20.41±7.15 | 7.5±7.86 | 3.78±7.49 | -0.39±7.24 | 13.74±6.99 | 19.19±7.52 | 10.64±2.31 |
| ABC | Climatology++ | 11.5±5.68 | 12.83±5.84 | 7.48±6.58 | 10.93±5.82 | 29.44±6.06 | 16.45±6.02 | 20.17±6.11 | 18.19±5.86 | 22.39±5.97 | 36.79±4.32 | 18.61±1.93 |
| | CFSv2++ | 34.46±4.77 | 43.23±5.63 | **13.67±5.27** | **32.3±5.41** | 37.36±5.18 | **43.06±4.79** | **26.1±5.29** | 26.93±6.4 | **33.85±5.32** | 30.77±5.91 | 32.2±1.74 |
| | Persistence++ | **41.21±3.96** | **47.39±4.86** | 12.48±4.78 | 27.99±5.7 | **37.44±4.89** | 41.03±5.11 | 24.48±4.95 | **32.64±6.16** | 18.51±6.01 | **40.53±5.51** | **32.4±1.72** |
| Learning | AutoKNN | 16.77±4.53 | 3.01±5.08 | 5.62±5.28 | 4.74±4.77 | 12.99±5.19 | 14.6±5.07 | 14.46±4.65 | 11.32±4.85 | 11.65±5.32 | 29.2±5.38 | 12.43±1.61 |
| | LocalBoosting | 10.51±5.65 | 17.85±5.14 | 5.12±3.99 | 15.5±5.34 | 20.68±4.03 | 17.94±5.17 | 1.99±5.67 | 18.95±4.69 | 17.46±5.18 | 17.88±5.48 | 14.44±1.63 |
| | MultiLLR | **34.05±5.15** | **31.03±5.95** | **9.57±4.75** | **22.7±5.94** | **29.87±5.92** | 23.93±5.57 | 18.85±5.64 | 20.68±5.78 | **29.98±5.46** | 24.16±6.31 | **24.5±1.8** |
| | Prophet | 18.31±4.37 | 14.51±5.05 | 3.86±4.15 | 14.3±4.41 | 14.8±4.94 | **27.74±5.03** | **26.6±5.11** | 25.76±6.84 | 20.76±5.14 | **35.74±3.6** | 20.21±1.57 |
| | Salient 2.0 | 6.29±6.46 | 2.31±6.3 | 0.01±6.17 | -5.73±6.59 | 15.13±5.68 | -4.32±6.57 | 23.01±6.03 | **28.83±6.02** | 14.54±6.08 | 32.73±6.21 | 11.24±2.03 |
| Ensembles | Uniform ABC | **40.16±4.4** | **45.49±5.27** | 12.4±5.5 | 30.33±5.84 | **39.06±5.18** | 42.23±5.15 | **28.85±4.96** | 29.12±6.1 | 30.31±5.58 | **37.34±5.66** | **33.55±1.72** |
| | Online ABC | 35.98±4.49 | 45.1±5.59 | **12.5±5.13** | **31.75±5.46** | 38.57±5.11 | **44.09±4.74** | 27.77±5.23 | **29.35±6.46** | **31.04±5.31** | 36.26±5.91 | 33.26±1.72 |
| | | | | | Temperature, weeks 5-6 | | | | | | | |
| Baselines | Deb. CFSv2 | **33.4±4.82** | **36.09±5.92** | **1.35±6.32** | 15.48±6.27 | **25.72±5.32** | **26.47±5.69** | **4.31±6.1** | **15.82±6.26** | **8.47±6.63** | **23.54±6.31** | **19.12±1.94** |
| | Persistence | 20.28±5.93 | -9.52±7.92 | 0.4±7.5 | **18.22±6.96** | 23.24±7.51 | -1.24±8.34 | -3.89±7.43 | -5.41±7.6 | 6.15±6.75 | 13.55±7.53 | 6.22±2.38 |
| ABC | Climatology++ | 11.5±5.68 | 12.83±5.84 | 7.48±6.58 | 10.93±5.82 | 29.44±6.06 | 16.45±6.02 | **22.79±5.83** | 18.19±5.86 | **22.39±5.97** | 36.79±4.32 | 18.87±1.92 |
| | CFSv2++ | 31.11±4.28 | 38.71±5.27 | **8.61±4.91** | **34.55±5.07** | **38.72±6.01** | **43.36±5.12** | 12.94±5.46 | 24.37±6.33 | 19.7±5.9 | 35.38±5.04 | **28.8±1.76** |
| | Persistence++ | **34.74±4.19** | **39.38±4.31** | 6.45±4.79 | 16.37±5.83 | 32.26±4.63 | 36.06±5.07 | 16.26±4.9 | **25.05±5.81** | 18.43±5.8 | **41.9±4.71** | 26.73±1.67 |
| Learning | AutoKNN | 4.73±4.61 | -0.32±4.79 | 8.06±4.63 | 5.8±4.68 | 11.43±4.93 | 6.28±4.29 | 9.89±3.57 | 8.21±4.67 | 4.44±5.19 | 27.12±5.43 | 8.56±1.52 |
| | LocalBoosting | 8.63±5.64 | 15.77±5.19 | 3.24±4.12 | 5.81±5.66 | **22.93±4.12** | 7.99±5.41 | 7.84±5.85 | 18.43±4.74 | 13.22±5.18 | 22.87±4.84 | 12.69±1.64 |
| | MultiLLR | **19.89±5.49** | **23.05±5.76** | 3.55±5.5 | **17.71±4.98** | 13.2±6.72 | 26.98±5.67 | 12.09±6.44 | 12.08±5.39 | 14.44±5.91 | 23.63±6.36 | 16.68±1.85 |
| | Prophet | 17.23±4.41 | 13.04±4.9 | 3.45±4.18 | 12.99±4.43 | 13.74±4.98 | 27.2±5.03 | **27.74±5.05** | 26.16±6.87 | **20.72±5.14** | **35.83±3.61** | **19.78±1.57** |
| | Salient 2.0 | 7.13±6.54 | -1.75±6.46 | **8.14±6.08** | -5.24±6.69 | 13.94±5.63 | -9.7±6.46 | 23.99±5.86 | **30.29±6.11** | 16.04±5.68 | 35.36±6.17 | 11.77±2.03 |
| Ensembles | Uniform ABC | **36.14±4.11** | **41.1±4.8** | **8.66±5.13** | 28.47±5.75 | 39.19±5.42 | 41.42±5.24 | **17.67±5.08** | **26.02±5.83** | **21.86±5.88** | **41.15±4.68** | **30.22±1.7** |
| | Online ABC | 33.3±4.15 | 40.1±5.18 | 7.82±4.86 | **30.41±5.23** | **39.61±5.79** | **42.12±5.07** | 15.76±5.31 | 25.86±6.16 | 21.32±5.91 | 40.74±4.78 | 29.76±1.72 |

Table 6: Average percentage skill when forecasting precipitation in the contiguous U.S. The best performing models within each group are shown in bold, while the best performing models overall are shown in green.

| Group | Model | 2011 | 2012 | 2013 | 2014 | 2015 | 2016 | 2017 | 2018 | 2019 | 2020 | Overall |
|---|---|---|---|---|---|---|---|---|---|---|---|---|
| | | | | | Precipitation, weeks 3-4 | | | | | | | |
| Baselines | Deb. CFSv2 | **15.26±3.38** | 8.99±3.52 | **5.96±3.72** | 5.69±3.88 | 3.69±3.74 | **2.19±3.61** | 0.34±4.05 | 4.01±3.44 | 3.76±3.08 | 7.19±3.76 | 5.77±1.11 |
| | Persistence | 14.24±3.03 | **9.61±3.48** | 5.96±3.07 | **6.46±3.24** | **7.2±3.63** | 1.3±2.83 | **13.46±3.89** | **6.72±2.76** | **8.43±3.27** | **10.32±2.81** | **8.31±1.0** |
| ABC | Climatology++ | 12.24±3.22 | **24.09±3.46** | 13.4±2.89 | 15.67±2.83 | 7.97±3.69 | **17.88±3.04** | 14.29±3.69 | **13.39±3.2** | 8.76±3.11 | 22.65±2.67 | 15.04±1.06 |
| | CFSv2++ | **24.58±3.4** | 19.08±2.81 | **17.44±3.11** | **16.15±2.86** | 7.98±3.5 | 12.42±3.08 | **14.75±3.79** | 13.42±3.18 | **13.09±4.11** | **23.05±2.74** | **16.21±1.04** |
| | Persistence++ | 21.98±2.17 | 14.03±2.36 | 16.63±2.96 | 13.9±2.78 | **9.38±3.46** | 8.12±2.6 | 5.1±3.38 | 13.33±3.06 | 12.89±2.74 | 17.48±2.74 | 13.38±0.89 |
| Learning | AutoKNN | 13.86±3.37 | 18.66±4.01 | 10.13±3.87 | 8.6±4.08 | 3.11±4.87 | 3.03±4.03 | 5.29±4.94 | -0.5±3.7 | -3.27±3.72 | 7.56±3.35 | 6.66±1.29 |
| | LocalBoosting | 6.32±2.88 | 15.96±2.16 | 15.73±2.54 | 7.7±2.82 | 4.27±2.83 | 2.64±2.63 | **11.77±3.52** | **13.79±2.4** | **10.68±2.47** | 19.46±2.48 | 10.82±0.88 |
| | MultiLLR | 12.44±3.12 | 4.87±3.46 | 14.72±3.11 | 10.74±3.54 | 5.47±3.97 | 5.75±3.18 | 8.49±3.52 | 7.83±2.43 | 8.28±3.25 | 16.16±2.23 | 9.49±1.02 |
| | Prophet | **18.23±2.77** | 12.42±3.04 | 14.29±2.41 | **15.85±2.14** | 5.33±3.13 | **13.49±2.58** | 11.31±3.34 | 13.19±2.79 | 10.77±2.65 | **20.01±2.46** | **13.51±0.9** |
| | Salient 2.0 | 17.06±4.21 | **20.38±3.88** | **17.82±4.1** | 11.24±4.06 | **5.95±5.5** | 3.0±4.14 | 5.45±5.33 | 2.44±3.89 | 2.28±4.19 | 14.99±3.35 | 10.11±1.37 |
| Ensembles | Uniform ABC | **26.72±3.12** | 23.2±2.64 | **19.9±3.37** | 19.79±2.7 | 11.21±3.47 | 15.46±2.98 | 15.01±3.75 | **16.49±3.25** | **14.98±3.73** | **25.16±2.6** | **18.84±0.99** |
| | Online ABC | 24.72±3.31 | **23.45±2.65** | 18.74±3.2 | **20.07±2.65** | **11.44±3.62** | **18.29±3.11** | **15.35±3.86** | 16.29±3.25 | 14.38±3.7 | 25.03±2.49 | 18.81±1.03 |
| | | | | | Precipitation, weeks 5-6 | | | | | | | |
| Baselines | Deb. CFSv2 | **11.73±3.16** | **10.98±3.22** | 8.35±3.71 | 1.03±3.35 | -4.91±3.37 | 0.98±3.46 | 2.77±3.58 | 4.24±3.21 | -1.3±3.37 | 8.73±3.7 | 4.28±1.06 |
| | Persistence | 6.85±3.87 | 6.96±3.42 | **12.34±3.26** | **4.32±3.2** | **3.83±3.61** | **1.87±3.05** | **10.9±3.5** | **8.73±2.91** | **5.62±3.22** | **13.04±2.77** | **7.41±1.02** |
| ABC | Climatology++ | 12.28±3.21 | **24.11±3.5** | 13.33±2.89 | **15.67±2.83** | 7.97±3.69 | **17.88±3.04** | **13.66±3.64** | 13.39±3.2 | 8.76±3.11 | **22.65±2.67** | 14.99±1.06 |
| | CFSv2++ | **23.11±3.33** | 18.75±2.81 | **22.87±2.96** | 14.88±2.59 | **8.42±3.33** | 13.44±3.44 | 7.74±4.31 | **15.34±3.21** | **13.4±3.7** | 22.17±2.67 | **16.11±1.04** |
| | Persistence++ | 17.44±2.65 | 9.98±2.81 | 16.9±2.61 | 7.68±2.76 | 2.44±3.38 | 3.83±2.83 | 3.6±3.58 | 11.26±2.42 | 7.6±2.6 | 16.26±2.22 | 9.77±0.89 |
| Learning | AutoKNN | 12.33±3.58 | 16.31±4.18 | 11.16±3.96 | 6.85±3.82 | 3.31±5.04 | 5.07±4.13 | 1.53±5.06 | -1.05±3.72 | -4.58±3.64 | 7.88±3.51 | 5.93±1.3 |
| | LocalBoosting | 9.81±2.98 | 9.26±2.38 | 13.24±2.34 | 4.53±3.04 | 5.31±2.96 | 6.7±2.7 | **10.25±3.7** | **13.55±2.78** | **10.31±2.71** | 14.3±2.44 | 9.72±0.9 |
| | MultiLLR | 12.42±3.58 | 5.99±3.05 | 10.69±2.76 | 7.46±3.39 | 0.63±4.03 | 1.97±3.32 | 10.03±3.54 | 8.46±2.79 | 6.83±3.13 | 15.5±2.37 | 7.97±1.03 |
| | Prophet | **17.92±2.78** | 12.41±3.04 | 14.74±2.42 | **15.86±2.14** | 5.44±3.1 | **13.54±2.61** | 10.06±3.45 | 13.08±2.79 | 10.75±2.65 | **19.94±2.48** | **13.41±0.91** |
| | Salient 2.0 | 17.06±4.18 | **20.26±3.86** | **17.73±4.03** | 11.48±4.03 | **5.91±5.47** | 2.77±4.09 | 5.1±5.29 | 2.06±3.99 | 2.1±4.19 | 14.86±3.41 | 9.99±1.37 |
| Ensembles | Uniform ABC | **25.31±3.11** | 23.05±2.64 | **23.45±2.82** | 17.89±2.62 | **9.17±3.4** | 16.63±3.57 | 11.23±4.0 | **16.8±3.08** | **13.93±3.34** | **25.24±2.37** | **18.35±0.98** |
| | Online ABC | 23.28±3.2 | **23.32±2.72** | 21.65±2.86 | **18.11±2.54** | 7.69±3.59 | **17.29±3.5** | **12.48±3.96** | 16.29±3.12 | 12.86±3.5 | 25.24±2.41 | 17.88±1.02 |

## C.4 Spatial Improvement over Mean Debiased CFSv2 RMSE

Figure 15 presents the spatial improvement of each model over debiased CFSv2 when predicting U.S. temperature across 2011–2020. For both the weeks 3-4 and the weeks 5-6 lead times, the ABC models uniformly improve over debiased CFSv2 and outperform both the baseline models and the state-of-the-art learning methods. The best overall performance at each lead time is obtained by the Online ABC ensemble.

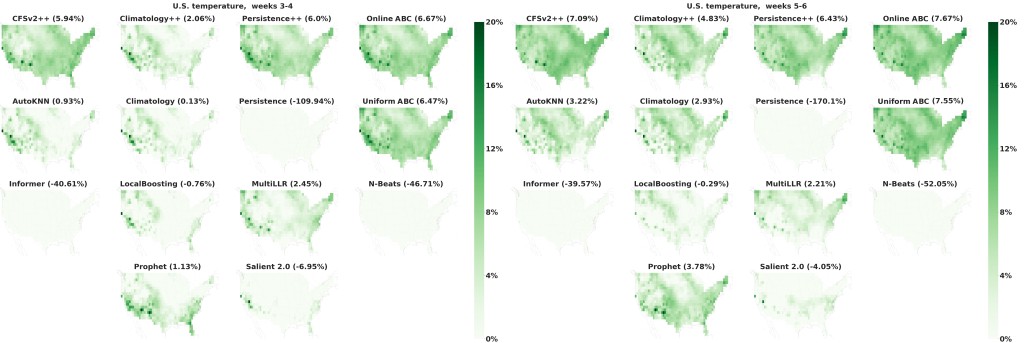

Figure 15: Percentage improvement over mean debiased CFSv2 RMSE when forecasting temperature in the contiguous U.S. over 2011–2020. White grid points indicate negative or 0% improvement, and the mean percentage improvement is given in parentheses.

Figure 16 displays the spatial improvement of each model over debiased CFSv2 when forecasting U.S. precipitation across 2011–2020. For precipitation, all models exhibit larger gains over debiased CFSv2, and all models, including Climatology, achieve larger improvements in the Western U.S. than in the Eastern U.S.

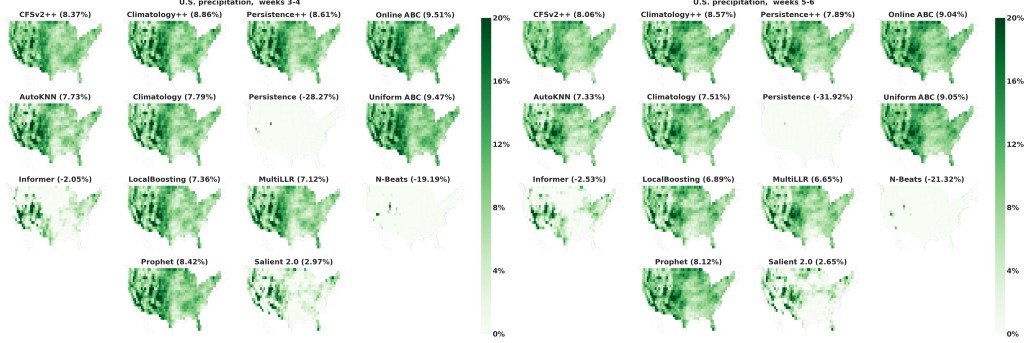

Figure 16: Percentage improvement over mean debiased CFSv2 RMSE when forecasting precipitation in the contiguous U.S. over 2011–2020. White grid points indicate negative or 0% improvement, and the mean percentage improvement is given in parentheses.

## C.5 Spatial Bias Maps

This section explores the spatial bias (the mean forecast minus the mean observation at each grid point in the contiguous U.S.) of each model across 2011–2020. The temperature maps of Figure 17 indicate a cold bias for most models over the southern half of the U.S. and an additional warm bias for several models in the center north. This warm bias is particularly pronounced for Salient 2.0. In precipitation maps of Figure 18, all models Salient 2.0 and AutoKNN show wet biases in the western half of the U.S. and dry biases in the eastern half. AutoKNN exhibits a dry bias extending from the Eastern U.S. to include the Northern U.S. as well, while Salient 2.0 displays a strong dry bias across the entire contiguous U.S. that is especially pronounced in the eastern half.

The Prophet model is noticeably less biased than the other evaluated models; however, this bias reduction does not immediately translate into improved performance, as Prophet is outperformed by ABC models with larger bias in all four tasks. This indicates that Prophet's reduced bias comes at a cost of unnecessarily high variance relative to the dominating ABC model forecasts.

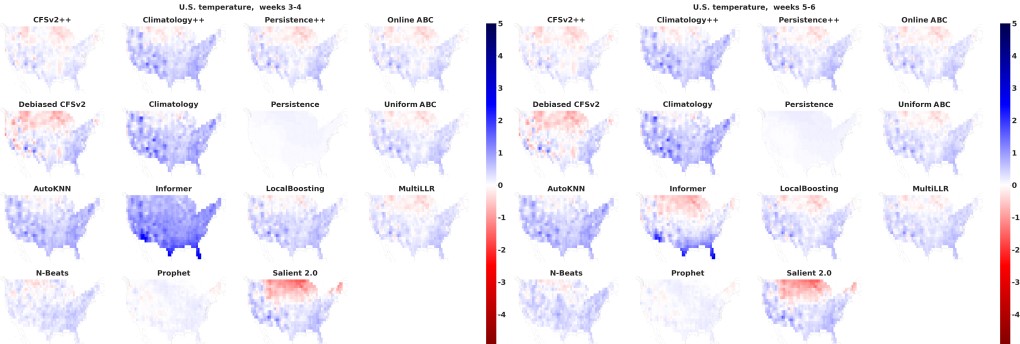

Figure 17: Model bias when forecasting temperature in the contiguous U.S. over 2011–2020.

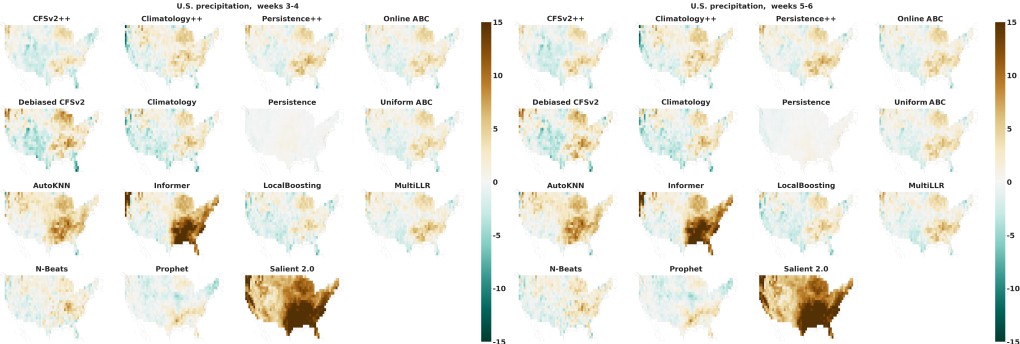

Figure 18: Model bias when forecasting precipitation in the contiguous U.S. over 2011–2020.

## C.6 GraphCast Comparison Details

To generate a weeks 3-4 GraphCast forecast of temperature and precipitation for a given target date, we

1. downloaded the pretrained `GraphCast - ERA5 1979-2017 - resolution 0.25 - pressure levels 37 - mesh 2to6 - precipitation input and output` model weights from `https://console.cloud.google.com/storage/browser/dm_graphcast`,

2. initialized the model with the three days of ECMWF Reanalysis v5 (ERA5) data preceding the associated issuance date, as described in `https://github.com/google-deepmind/graphcast`,

3. repeatedly ran the model (using autoregressive rollout as in `https://github.com/google-deepmind/graphcast/blob/main/graphcast_demo.ipynb`) for the maximum supported number of time steps (12) and reinitialized the model with its own forecasted output until 30 days of forecasts had been produced,

4. aggregated the `2m_temperature` and `total_precipitation_6hr` forecasts over days 14-28,

5. regridded the aggregated forecasts to our standard $1 \times 1°$ grid using `xarray interp_like` [22], and

6. converted temperature to degrees Celsius and precipitation to millimeters.

Since GraphCast was trained on data through 2017, Table 7 summarizes the performance of GraphCast across the post-training years 2018–2020. Similarly to the deep learning models previously evaluated in Table 1, GraphCast outperforms debiased CFSv2 in terms of skill, underperforms debiased CFSv2 in terms of RMSE, and strongly underperforms the ABC ensemble models in both metrics when forecasting either temperature or precipitation in weeks 3-4.

Table 7: Average percentage skill and percentage improvement over mean debiased CFSv2 RMSE across 2018–2020 in the contiguous U.S. along with a 95% bootstrap confidence interval. The best performing model overall is shown in green.

| | | RMSE % IMPROVEMENT | | AVERAGE % SKILL | |
| | | WEEKS 3-4 | | | |
| GROUP | MODEL | TEMPERATURE | PRECIPITATION | TEMPERATURE | PRECIPITATION |
|---|---|---|---|---|---|
| BASELINES | DEB. CFSV2 | – | – | 19.98±4.84 | 3.12±2.99 |
| LEARNING | GRAPHCAST | −34.96±7.46 | −34.26±7.57 | 20.17±4.70 | 5.55±2.81 |
| ENSEMBLES | UNIFORM ABC | **9.70±2.19** | **8.41±1.18** | 34.37±4.58 | **18.34±2.59** |
| | ONLINE ABC | 9.37±2.07 | 8.37±1.25 | **34.67±4.71** | 18.07±2.76 |

Figure 19 displays the spatial improvement of GraphCast over debiased CFSv2 when forecasting U.S. temperature and precipitation across 2018–2020. Unlike the ensemble ABC models, GraphCast seldom improves over the debiased CFSv2 baseline.

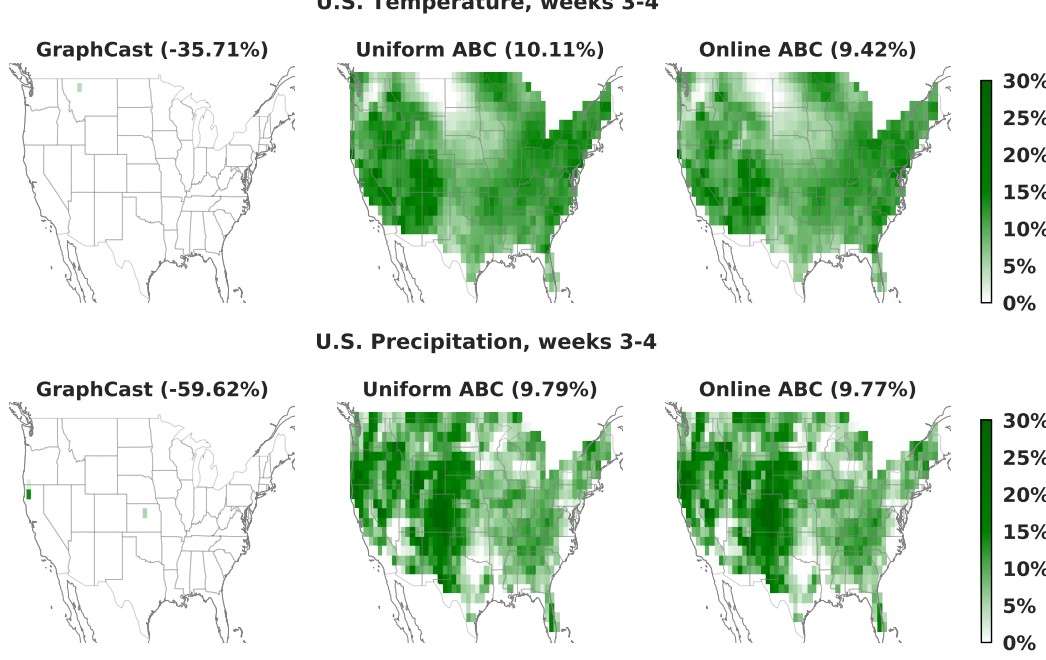

Figure 19: Percentage improvement over mean debiased CFSv2 RMSE when forecasting temperature or precipitation in the contiguous U.S. over 2018–2020. White grid points indicate negative or 0% improvement, and the mean percentage improvement is given in parentheses.

Figure 20 compares the spatial bias (the mean forecast minus the mean observation at each grid point in the contiguous U.S.) of GraphCast versus the ABC ensemble models across 2018–2020. For

precipitation, GraphCast has a substantially drier bias than the ABC models throughout most of the contiguous U.S. For temperature, GraphCast has a warmer bias in the West.

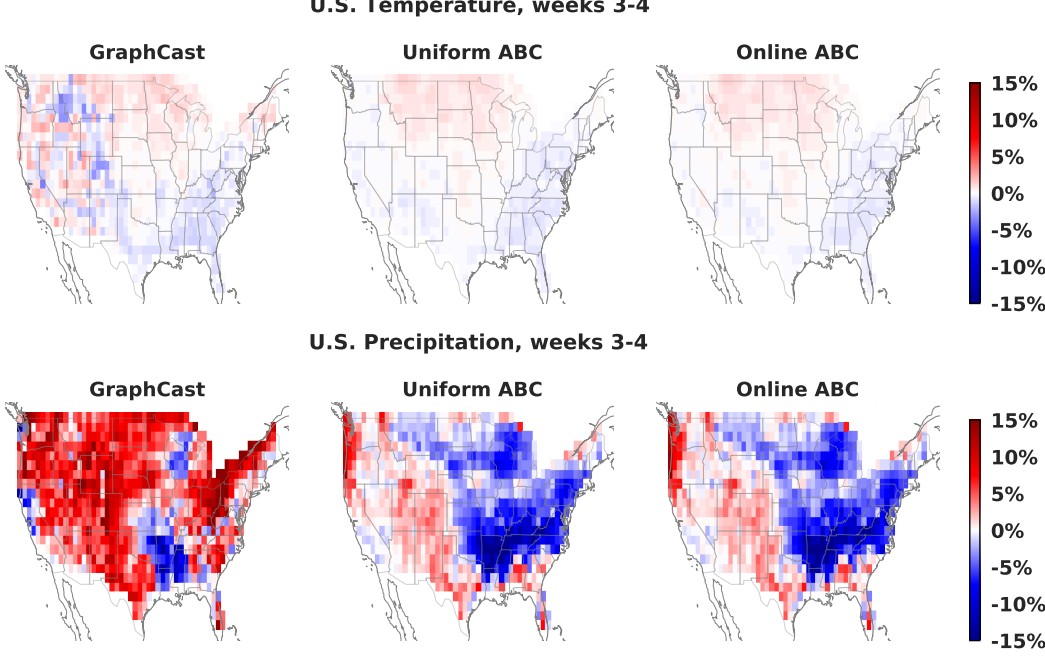

Figure 20: Bias of GraphCast vs. ABC ensembles when forecasting temperature or precipitation in the contiguous U.S. over 2018–2020.

## C.7   Western U.S. Competition Results

Table 8 displays the performance improvement over the baseline debiased CFSv2 in the Western U.S., over 26 dates in 2019-2020. This region was the focus of the Subseasonal Forecast Rodeo I, and we include the performance of the three best contestants in the table. The ABC ensembles were either very competitive or better than the other contestants.

Table 8: Percentage improvement over mean debiased CFSv2 RMSE over 26 contest dates (2019-2020) in the Western U.S. The best performing models within each class of models are shown in bold, while the best performing models overall are shown in green.

| Group | Model | Temp. weeks 3-4 | Temp. weeks 5-6 | Precip. weeks 3-4 | Precip. weeks 5-6 |
|---|---|---|---|---|---|
| Contest baselines | Salient | − | − | 11.10 | 7.02 |
| | Climatology | 10.22 | −0.76 | 5.82 | 2.25 |
| Contestants | $1^{st}$ place | **17.12** | **8.47** | **11.54** | **8.63** |
| | $2^{nd}$ place | 16.67 | 7.04 | 11.10 | 8.03 |
| | $3^{rd}$ place | 15.47 | 6.90 | 10.62 | 7.94 |
| Learning | AutoKNN | 13.09 | 2.90 | 7.50 | 3.05 |
| | LocalBoosting | 12.85 | 4.09 | 7.25 | 3.71 |
| | MultiLLR | 9.54 | 1.12 | 8.95 | 4.58 |
| | Prophet | **15.68** | **6.86** | 6.88 | 3.40 |
| | Salient 2.0 | 11.15 | 2.91 | **12.65** | **8.56** |
| ABC | Climatology++ | 15.54 | 6.43 | 8.35 | 4.69 |
| | CFSv2++ | 6.67 | **9.26** | **8.70** | **5.51** |
| | Persistence++ | **16.59** | 8.27 | 8.20 | 4.51 |
| Ensembles | Uniform ABC | 14.96 | **9.58** | 9.31 | 5.89 |
| | Uniform ABC + Learning | 15.89 | 8.79 | 10.43 | 6.79 |
| | Online ABC | **16.71** | 8.70 | 8.85 | 5.19 |
| | Online ABC + Learning | 14.70 | 7.97 | **12.52** | **8.18** |

## C.8 Salient 2.0 Dry Bias

We hypothesized that the exceptional Western U.S. contest performance of Salient 2.0 in Table 8 was due in part to the dry bias observed in Figure 18, as 2020 was an unusually dry year in the Western U.S. To explore this hypothesis, we focus on forecasting precipitation weeks 3-4 in the Western U.S. and display in Figure 21 (left) the percentage improvement of Salient 2.0 and CFSv2++ over debiased CFSv2 alongside the inverse total precipitation each year in 2011–2020. As anticipated, the steep decrease in cumulative precipitation for 2020 is accompanied by a steep increase in Salient 2.0 predictive accuracy. In addition, the rises and falls in total precipitation track the accuracy of Salient 2.0 well but appear largely unassociated with the performance curve of other accurate models like CFSv2++. The scatter plots and best-fit lines of Figure 21 (right) paint a similar picture. Salient 2.0 exhibits a distinctly negative correlation between percentage improvement and total precipitation in the Western U.S., while this relationship is absent for other accurate models like CFSv2++.

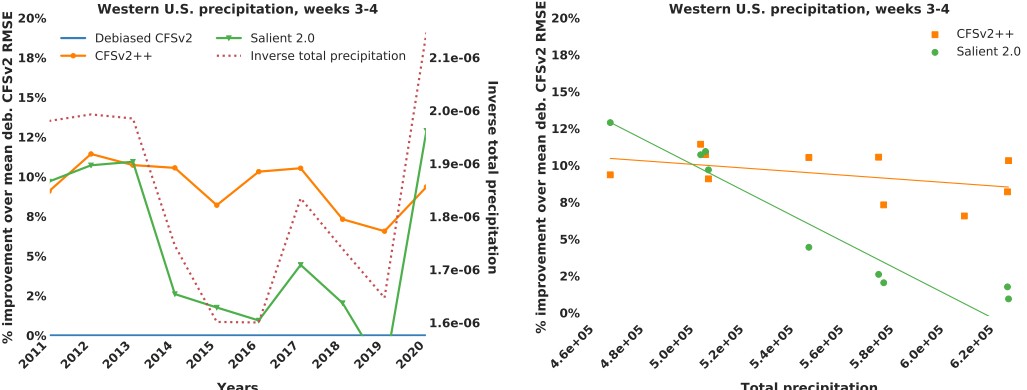

Figure 21: Temporal plot (left) and scatter plot (right) of yearly total precipitation and percentage improvement over mean debiased CFSv2 RMSE in the Western U.S. across 2011–2020.

The developers of the Rodeo I Salient model attribute this dry bias to the log-normal distribution of precipitation, which leads to more frequent anomalously dry conditions than anomalously wet conditions and in turn encourages drier model forecasts [58].

More recent versions of the Salient model mitigate this bias by training on seasonal anomalies in place of raw temperature and precipitation values [58].

