# OpenReview forum: "SubseasonalClimateUSA: A Dataset for Subseasonal Forecasting and Benchmarking"
_NeurIPS.cc/2023/Track/Datasets_and_Benchmarks — NeurIPS 2023 Datasets and Benchmarks Poster_

### Official Review · Reviewer_SYV5 · 2023-07-21
**SubseasonalClimateUSA: A Dataset for Subseasonal Forecasting**

**Rating:** 6
**Confidence:** 3
**Correctness:** Yes
**Clarity:** Yes

**Strengths:**

1. Dataset covers seven meteorological data sources for collecting the data and provides machine-learning-ready Python Pandas DataFrames.
2. The coverage is over the contiguous USA which is an improvement over the SubseasonalRodeo dataset.
3. Clear description of the dataset and any data augmentations.
4. Extensive experiments covering several benchmarks.

**Additional Feedback:**

NA

**Documentation:**

The documentation (supplementary material) and the GitHub page are clearly written. The GitHub repository seems to be actively maintained.

**Limitations:**

1. Major concern for me is the lack of novelty of this new dataset in comparison with the SubseasonalRodeo dataset. It is incremental work.
2. With a little web search I stumbled upon an earlier version of this paper (https://arxiv.org/pdf/2109.10399v1.pdf) and I noticed that the benchmark section in the current submission does not include any new benchmarks compared to the earlier version of this paper which was submitted 2 years back.

**Opportunities For Improvement:**

NA

**Relation To Prior Work:**

Incremental

**Summary And Contributions:**

The work introduces the SubseasonalClimateUSA dataset for training and benchmarking subseasonal forecasting models in the United States. The current dataset is a successor to the SubseasonalRodeo dataset, in that it provides new variables, expanded domain (Western USA vs the entire USA), and is regularly updated (whereas the Rodeo dataset is a static snapshot).

---

> ### Author Response · Authors · 2023-08-13
> **Novelty**
>
> We apologize for not better emphasizing the important differences between the SubseasonalClimateUSA dataset and the SubseasonalRodeo dataset.  In the revision we will highlight that:
> - SubseasonalClimateUSA is a modern, regularly updated resource while SubseasonalRodeo is a static snapshot with no data after 2018.
> - SubseasonalClimateUSA targets the entire contiguous US, while SubseasonalRodeo is restricted to the Western US.
> - SubseasonalClimateUSA provides granular (daily and subweekly) forecasts from the leading subseasonal dynamical model (ECMWF) and from seven other operational dynamical models in the SubX consortium. SubseasonalRodeo provides no forecasts from the leading subseasonal dynamical model (ECMWF) and includes only coarse-grained (monthly) forecasts from the North American Multi-Model Ensemble, with limited utility for weekly or biweekly forecasting.
>
> In addition to the dataset release, our submission provides an extensive and novel set of benchmark experiments comparing the skill and RMSE of 17 subseasonal forecasting methods on four core subseasonal forecasting tasks  across the contiguous US. Two of these models (Salient 2.0, the best-performing neural network method, and LocalBoosting) were novel creations of this work, and ten required new subseasonal forecasting implementations that we have now released through the new subseasonal_toolkit package.

---

> ### Author Response · Authors · 2023-08-13
> **Benchmarks**
>
> Our experiments benchmarking 17 models included all state-of-the-art subseasonal forecasting models of which we were aware, but please let us know if we have missed any relevant candidates, as we would be happy to include them in our evaluation. Following the recommendation of kSL6, we intend to evaluate the recent deep learning methods PanguWeather and Graphcast [Bi et al., 2023, Lam et al., 2023] as two additional benchmarks. Note that neither was designed for or tested on subseasonal timescales, but both have shown promising performance on shorter time horizons. In addition, we will highlight that ABC [Mouatadid et al., 2023] was recently shown to outperform three additional hybrid physics+learning models including the recent neural network debiasing scheme of Fan et al., 2021.
>
> ----------
> References
>
> - K. Bi, L. Xie, H. Zhang, X. Chen, X. Gu, and Q. Tian. Accurate medium-range global weather forecasting with 3D neural networks. Nature, pp.1-6, 2023.
>
> - R. Lam, A. Sanchez-Gonzalez, M. Willson, P. Wirnsberger, M. Fortunato, A. Pritzel, S. Ravuri, S., T. Ewalds, F. Alet, Z. Eaton-Rosen, and W. Hu.  GraphCast: Learning skillful medium-range global weather forecasting. arXiv preprint arXiv:2212.12794. 2022.
>
> - S. Mouatadid, P. Orenstein, G. Flaspohler, J. Cohen, M. Oprescu, E. Fraenkel, L. Mackey. Adaptive bias correction for improved subseasonal forecasting. Nature Communications. 2023 Jun 15;14(1):3482.

---

> ### Comment · Reviewer_SYV5 · 2023-08-18
> **Response to the authors' comments**
>
> Novelty:
> 1. The first two points were clear from your current manuscript. I might have missed the 3rd point during my initial review of your manuscript.
> 2. I am aware that your data is regularly updated and hence serves as a better baseline than the SubseasonalRodeo in future works.
> 3. It still does not invalidate my concern regarding the lack of novelty towards the "dataset" component of the "NeurIPS Track Datasets and Benchmark".
> 4. However, with the genuine public interest and extensive coverage with regular updates provided by your dataset, I am going to increase my score by 1. Beyond this point, I will leave it up to the area chairs to decide if my lack of novelty concerns can be overlooked to accept your paper.
>
> Benchmarks:
> 1. the addition of the PanguWeather and Graphcast model results would definitely bring your benchmark component up to the present year.
> 2. More discussion as you highlighted in your rebuttal to other reviewers about why the simple baselines do better than the others would definitely improve your revision.
> 3. If authors think that extVAE (https://arxiv.org/pdf/2307.08079.pdf) could be applied to their subseasonal forecasting task then I would like to see it added into your benchmark suite. However, I understand that this work is unpublished so this is just a request, and if authors chose not to do it, then that is fine.
> 4. The benchmark component of this paper is strong and assuming that authors incorporate reviewers' suggestions into the final version, I am going to preemptively increase my score by 1.

---

### Official Review · Reviewer_kSL6 · 2023-07-21
**Review for subseasonal benchmark dataset**

**Rating:** 6
**Confidence:** 2
**Correctness:** Yes
**Clarity:** Yes, to some extent (see questions)

**Strengths:**

The main strength is in the introduction of a subseasonal dataset that focuses on longer time scales unlike current benchmarks that focus on medium-range weather prediction (up to 2 weeks) and the evaluation of models on this dataset. Demonstrating skill on this longer timescale is very important to advance ML models for weather/climate and hence this dataset can prove to be very useful.

**Additional Feedback:**

1. Can the authors clarify what the ground truth is when computing metrics? Is it just the single dynamical model CFSv2? What about the other models?
2. Can the authors clarify the training setup (which years for training, validation, etc), what they are trying to predict (is it an autoregressive forecast over 6 weeks of the state variables?). While details can be in the appendix, it will be more clear if in the main paper, the goal is clearly stated as well as what the truth is.
3. I am a little confused about this statement in the introduction: "there are no general datasets targeting subseasonal scale for the US". If I downloaded ERA5 from ECMWF at 0.25 deg, why does it not represent a subseasonal scale dataset? For the US, can just focus on this specific region. Is it just the addition of several more variables relating to subseasonal processes as well as other dynamical model outputs? Please clarify.
4. It would be interesting to see if any of the SOTA weather forecasting emulators today (PanguWeather, Graphcast, Fourcastnet etc) have subseasonal scale. Have you considered testing these medium-range forecasting models on larger time scales to provide a benchmark? One issue is that they do run at higher 0.25 deg resolution than this paper.

**Documentation:**

Yes

**Limitations:**

- The authors talk about resolution limitations as well as region limitations in the paper

**Opportunities For Improvement:**

- The resolution seems quite coarse for any extreme event prediction.  Weather forecasting datasets for benchmarking ML models today are from ERA5 at 0.25 deg (compared to 1.5 or 1 deg here). Regional forecasting typically would go even higher than this.
- ACC and RMSE are averaged metrics. For extreme events, other percentile related metrics or PDFs might prove more useful

**Relation To Prior Work:**

Yes

**Summary And Contributions:**

The authors present a comprehensive dataset over the US for subseasonal (2 to 6 weeks) forecasting of key atmospheric variables that are of relevance in this temporal scale. They then benchmark various types of models that include simple baselines as well as ML models to show their performance in forecasting the above variables.

---

> ### Author Response · Authors · 2023-08-13
> **Resolution**
>
> Thank you for raising this potential point of confusion. We will clarify in the revision that
> 1. Our ground-truth measurements are in fact downloaded on 0.25 or 0.5 deg grids (depending on the variable), and this granular resolution is also available through the SubseasonalClimateUSA dataset.
> 1. However, the spatial bottleneck is provided by the operational dynamical models which only provide forecasts at 1.5 deg resolution (for ECMWF) or 1 deg resolution (for the SubX).
> 1. Relatedly, while a finer spatial granularity is more common for shorter-term weather forecasting, 1.5 and 1 deg resolutions are standard for subseasonal forecasting. For example, the recent subseasonal AI challenge hosted by the World Meteorological Organization (https://s2s-ai-challenge.github.io/) required participants to produce forecasts on a 1.5 deg grid, while the two preceding Subseasonal Forecast Rodeos, hosted by the US Bureau of Reclamation and NOAA enforced a 1 degree grid.

---

> ### Author Response · Authors · 2023-08-13
> **Alternative metrics**
>
> Thank you for this recommendation. Our benchmarking, like much past work on subseasonal forecasting [Monhart et al., 2018, Baker et al., 2019, Li et al., 2019, Fan et al., 2021,Kim et al., 2021], focuses on improved deterministic forecasting, that is predicting a point estimate of temperature or precipitation rather than a distribution over those quantities. For this end, we adopted the exact evaluation protocols of the two Subseasonal Climate Forecast Rodeo competitions, run by the US Bureau of Reclamation in partnership with the National Oceanic and Atmospheric Administration, U.S. Geological Survey, U.S. Army Corps of Engineers, and California Department of Water Resources [Nowak et al., 2017, Nowak et al., 2020]. Nevertheless, we agree that probabilistic forecasting (e.g., predicting the future probability of climatological quantiles) is an important complementary paradigm that can also be evaluated using SubseasonalClimateUSA. In fact, benchmarking probabilistic forecasting atop the SubseasonalClimateUSA dataset is an important next step for our team. We have already included some variables more suitable for probabilistic forecasting in the dataset, and we include the standard probabilistic forecasting metrics continuous ranked probability scores (CRPS) and Brier skill scores (BSS) as evaluation options in the subseasonal_toolkit Python package.
>
>
> ------
> References
>
> - S. A. Baker, A. W. Wood, and B. Rajagopalan. Developing subseasonal to seasonal climate forecast products for hydrology and water management. JAWRA Journal of the American Water Resources Association, 55(4):1024–1037, 2019.
>
> - Y. Fan, V. Krasnopolsky, H. van den Dool, C.-Y. Wu, and J. Gottschalck. Using artificial neural networks to improve CFS week 3-4 precipitation and 2-meter air temperature forecasts. Weather and Forecasting, 2021. doi: 10.1175/WAF-D-20-0014.1. URL https://journals.ametsoc.org/view/journals/wefo/aop/WAF-D-20-0014.1/WAF-D-20-0014.1.xml.
>
> -  H. Kim, Y. Ham, Y. Joo, and S. Son. Deep learning for bias correction of mjo prediction. Nature communications, 12(1):1–7, 2021.
>
> -  W. Li, J. Chen, L. Li, H. Chen, B. Liu, C.-Y. Xu, and X. Li. Evaluation and bias correction of S2S precipitation for hydrological extremes. Journal of Hydrometeorology, 20(9):1887–1906, 2019.
>
> -  S. Monhart, C. Spirig, J. Bhend, K. Bogner, C. Sch¨ar, and M. A. Liniger. Skill of subseasonal forecasts in Europe: Effect of bias correction and downscaling using surface observations. Journal of Geophysical Research: Atmospheres, 123(15):7999–8016, 2018.
>
> -  K. Nowak, R. Webb, R. Cifelli, and L. Brekke. Sub-seasonal climate forecast rodeo. In 2017 AGU Fall Meeting, pages 11–15, New Orleans, LA, 2017.
>
> -  K. Nowak, I.M Ferguson, J. Beardsley, and L. D Brekke. Enhancing western united states sub-seasonal forecasts: Forecast rodeo prize competition series. In AGU Fall Meeting 2020. AGU, 2020.
>
> - S. Mouatadid, P. Orenstein, G. Flaspohler, J. Cohen, M. Oprescu, E. Fraenkel, L. Mackey. Adaptive bias correction for improved subseasonal forecasting. Nature Communications. 2023 Jun 15;14(1):3482.

---

> ### Author Response · Authors · 2023-08-13
> **Ground truth**
>
> We apologize for the confusion and will clarify in lines 97-100 and 137-139 of the revision that the ground truth consists of 14 day-averaged temperature measurements and 14 day-accumulated precipitation measurements that were obtained from the NOAA Climate Prediction Center (CPC) Global Daily Gridded Temperature (ftp://ftp.cdc.noaa.gov/Datasets/cpc_global_temp/) and Global Daily Unified Gauge-Based Analysis of Precipitation (ftp://ftp.cdc.noaa.gov/Datasets/cpc_global_precip/.NOAA/OAR/ESRL) reanalysis datasets. We use this ground truth data to evaluate all models (including CFSv2) mentioned in our paper.

---

> ### Author Response · Authors · 2023-08-13
> **Training setup**
>
> Thank you for your suggestion; we agree that it would be helpful to bring additional information about the training setup from the appendix into the main text, and we will do so in the final version.
>
> For all models in our paper, we consider four target subseasonal prediction tasks at each grid point in the contiguous US: forecasting (i) average temperature in weeks 3-4, (ii) average temperature in weeks 5-6, (iii) total precipitation in weeks 3-4, and (iv) total precipitation in weeks 5-6 (see lines 137-139). For each target variable, individual models general direct forecasts (as opposed to iterative forecasts that are rolled forward). Our test set consists of every Wednesday in the ten-year period from 2011 to 2020 inclusive. All of our models adopt a progressive training and validation protocol in which, to produce a forecast for a given target date, a model can be (re)trained and (re)tuned using any data fully observable as of the associated forecast issuance date (i.e., 14 days prior for weeks 3-4 and 28 days prior for weeks 5-6). Most models train on all available data (subject to the progressive training and validation constraint) and tune hyperparameters using a second round of progressive evaluation over the prior three years. Appendix B contains the specific implementation details for each model, including training, validation (hyperparameter tuning), and testing details, and we would be happy to move any of this information into the main paper.

---

> ### Author Response · Authors · 2023-08-13
> **Beyond ERA5**
>
> We appreciate the chance to clarify this issue. ERA5 is a valuable data source, but other predictive features not captured by ERA5 are also highly relevant to the subseasonal prediction problem.  These include, for example, the state of the Madden-Julian Oscillation, the multivariate El Nino-Southern Oscillation index, elevation, climate classification, and operational dynamical model forecasts. The SubseasonalClimateUSA dataset weaves these diverse features from diverse data sources together into a standardized machine-learning ready dataset.

---

> ### Author Response · Authors · 2023-08-13
> **Evaluating weather forecasters on subseasonal timescales**
>
> Thank you for this excellent suggestion.  We had not thought to evaluate these recent weather models as they were designed for much shorter timescales (PanguWeather for instance was developed for 1 hour to 168 hour forecasts). However, an initial investigation suggests that they can be pushed to generate subseasonal forecasts, so we will gladly add them to our list of benchmarks. The work of Bi et al. (2023) demonstrated that the performance of FourcastNet is typically dominated by both PanguWeather and the IFS dynamical model from ECMWF, so we will focus our efforts on benchmarking PanguWeather and Graphcast. Notably, both models were trained on data from 1979–2017, and PanguWeather was additionally tuned on 2019, so the number of evaluation years will necessarily be limited.
>
>
> --------
> References
>
> - K. Bi, L. Xie, H. Zhang, X. Chen, X. Gu, and Q. Tian. Accurate medium-range global weather forecasting with 3D neural networks. Nature, pp.1-6, 2023.

---

> > ### Comment · Reviewer_kSL6 · 2023-08-29
> >
> > Many thanks to the authors for responding to all my comments. I raise my score to 6.

---

### Official Review · Reviewer_UtoY · 2023-07-21
**Comprehensive subseasonal dataset with some benchmarking concerns**

**Rating:** 7
**Confidence:** 3
**Clarity:** Clarity is fine.

**Strengths:**

1. Subseasonal forecasting is critical for a number of social and economic areas, with significant benefits to improved forecasts.
2. This is a comprehensive dataset which aggregates data from a variety of sources.
3. A large number of benchmarks are conducted, as is a comparison with SOTA competition winners.

**Additional Feedback:**

This seems like a quite useful dataset, but some details need to be ironed out.

**Correctness:**

I do not see any issues with the dataset as constructed. However, the benchmark results for learned methods do not include error bars or other measures of variance.

**Documentation:**

The dataset and baselines are well-documented and available. However, some details of the maintenance and updates are unclear. For example, the paper says that the dataset will be regularly updated, but I do not see any documentation on releases or changes to the dataset.

**Ethics:**

No ethical issues.

**Limitations:**

The details of the dataset and limitations are adequately discussed.

**Opportunities For Improvement:**

1. A number of the data sources cover different spans of time and have different spatial resolutions. How is this all standardized within the dataset?
2. Are there standard or recommended train/validation/test splits for the data? This would be beneficial for comparisons across papers.
3. Learned benchmark results should include error bars or some other measure of variance.
4. It is surprising that relatively simple models appear to be SotA NWP approaches. Do you have any insight into why this is?
5. One thing that was not clear to me is what the paper aims to have the community do with the dataset. Where should the community go? What benchmarks should users target?

-----

The authors have addressed much of my concerns in their response and I have raised my score accordingly.

**Relation To Prior Work:**

Prior work is well-discussed and, in some relevant cases, directly used or compared with.

**Summary And Contributions:**

The paper introduces a subseasonal dataset covering the contiguous United States, using data aggregated from a diverse set of sources. A number of models are used in benchmarking results for temperature and precipitation at two different lead times, and a comparison with the winners of the Subseasonal Climate Forecast Rodeo II competition is conducted. Data and model implementations are available via Python packages.

---

> ### Author Response · Authors · 2023-08-13
> **Standardization**
>
> We apologize for the confusion regarding this matter. In the revision, we will (1) highlight that complete details concerning standardization are provided in the dataset documentation https://github.com/microsoft/subseasonal_data/blob/main/DATA.md and (2) augment the standardization details in Section 2 (lines 91-95) with the most useful details. Temporal and spatiotemporal variables arising from daily data sources were derived by averaging input values over each 14-day period (except for precipitation which is summed rather than averaged), and spatial and spatiotemporal variables were derived by interpolating input data to a 1° × 1° latitude-longitude grid using the Climate Data Operators operator remapdis (distance-weighted average interpolation) with target grid r360x181 and retaining only the grid points belonging to the contiguous United States. To accommodate ECMWF forecasts, which were only made available at a 1.5ºx1.5º resolution, we also include temperature, precipitation, ECMWF, and SubX forecasts at a 1.5ºx1.5º resolution. All of our data is passed through a common pre-processing pipeline, resulting in standardized dataframes.

---

> ### Author Response · Authors · 2023-08-13
> **Recommended train/validation/test splits**
>
> Thank you for this excellent recommendation. In the revision, we will recommend adopting the Wednesdays of each complete year from 2011 onward as a standard test set when evaluating year by year (as in Fig. 3) and the Wednesdays from the ten year period 2011-2020 as a standard test set when comparing with the overall (Tab. 1) or seasonal (Fig. 3) performance reported in this work. We will also recommend a progressive training and validation protocol in which, to produce a forecast for a given target date, a model can be (re)trained and (re)tuned using any data fully observable on the associated forecast issuance date (i.e., 14 days prior for weeks 3-4 and 28 days prior for weeks 5-6). All of the models evaluated in our submission respect this protocol. In addition, the SubseasonalClimateUSA dataset provides convenient combination dataframes containing target variables associated with predictive features lagged by an appropriate amount to ensure that each predictive feature was fully observed on the issuance date associated with the target date.

---

> ### Author Response · Authors · 2023-08-13
> **Measures of variance**
>
> Thank you for this recommendation which we will incorporate into the revision.

---

> ### Author Response · Authors · 2023-08-13
> **Success of relatively simple models**
>
> Thank you for this question; surprising results like these are exactly why we believe it is so important to establish a standardized dataset for benchmarking subseasonal forecasting methods. We believe the answer to this question is multifaceted. First, even the leading physics-based dynamical models are subject to inaccuracies due to inexact measurement, incomplete representation of the environment, imperfect simulation, and chaos. Second, many of the more elaborate machine learning models studied in this work appear to be prone to overfitting in the presence of the relatively high noise levels of subseasonal forecasting. Third, the best-performing models, while relatively simple, are hybrid physics+learning models designed to leverage the strengths of an underlying dynamical model while simultaneously enhancing its predictive skill by reducing its systemic bias. We will add more discussion related to this point in the final version.

---

> ### Author Response · Authors · 2023-08-13
> **Directives for the community**
>
> We apologize for the lack of clarity surrounding these important questions. Our first aim is to facilitate the development of skillful learning-based subseasonal forecasting models by providing a comprehensive, standardized, and machine-learning-friendly dataset. Our second is to provide standardized benchmarks for subseasonal forecasting progress. To this end, we define four core subseasonal prediction tasks that users can use as benchmarking targets: forecasting (i) temperature in weeks 3-4, (ii) temperature in weeks 5-6, (iii) precipitation in weeks 3-4, and (iv) precipitation in weeks 5-6. Significant advances in any of these tasks would have significant implications for the allocation of water resources, agricultural production, and disaster relief.

---

> ### Author Response · Authors · 2023-08-13
> **Regular updates**
>
> Thank you for raising this point. The dataset update pipeline (responsible for downloading new data and integrating it into the processed dataframes) runs daily, and we will post the up-to-date update schedule on the subseasonal_data repository site.

---

> > ### Comment · Reviewer_UtoY · 2023-08-13
> > **Response**
> >
> > Thank you for your response and clarifying these points. This addressed a lot of my concerns. I intend to raise my score to a 7 in light of this. (Unfortunately, it appears that editing has not been enabled for us, so I cannot update my review at this time.)

---

> > > ### Author Response · Authors · 2023-08-14
> > > **Review editing is now enabled**
> > >
> > > Thank you for your encouraging response!  We spoke with the Datasets and Benchmarks chairs, and review editing should now be enabled.

---

### Official Review · Reviewer_sMGZ · 2023-07-27
**Review of SubseasonalClimateUSA dataset**

**Rating:** 9
**Confidence:** 4
**Correctness:** As far as I can judge, yes they are.

**Strengths:**

See also above - strong contribution on an important topic, where ML has great potential. This is a task that should be tackled in tandem by meteorologists and MLers and this dataset will help both communities to streamline their work. Suggested extensions to be build on this paper show a clear path forward to extend the dataset to global scale.

The work is also highly valuable from the ethical and social implications standpoints.

**Additional Feedback:**

N/A

**Clarity:**

Yes, clear language. Clarity issues mostly refer to additionally needed details, see above.

**Documentation:**

As far as I had time to check, this looks fine. Some additions in text are needed to support reproducibility, see above.

**Ethics:**

No ethical concerns.

**Limitations:**

My main concern is a more detailed discussion of why the ABC methods work best (and how they actually work so well) plus additional info on the interpolation methods to create the dataset. Otherwise, the limitations are discussed well in a separate subsection.

**Opportunities For Improvement:**

My suggestions mostly concern clarity - not in terms of writing style but in terms of additional detailed information. These points include:
- line 2: you here highlight the importance for 'climate adaptation', but it is not clear to me how immediate that link really is. In my mind, this refers to far longer timescales than 'subseasonal', so maybe best to remove this?
- without having worked with the particular datasets used here, it appears to me that the homogenization of the various measurements and model datasets must have been a key issue, which might also have substantial impacts on the final dataset the ML models are trained on. More details can certainly not harm to make future revisions comparable, in particular if other groups might want to add data and/or reproduce your results. For example, this includes the chosen time resolution. In line 93 you mention that you work with 14-day periods unless otherwise mentioned, but then a lot of the Data Features (lines 96 cont.) appear to be provided daily? Are the 14-days means continuously shifted, or discrete periods starting on a certain day in the 1980s or so? Actually, on that same note, in a quick second scan through the paper I cannot find which period is covered by your Dataset (sometimes 2010, sometimes 2020 is the end date?). It might be best to highlight this information prominently in the abstract and in one of the figures.
- could you birefly discuss again the advantages of merging observational and model data, despite their heterogeneity? More data doesn't always help, if the data e..g comes from a low-quality model, doesn't it?
- in line 93, you mention interpolation to a 1x1 degree grid and you also highlight issues related to various choices for interpolation methods under Dataset Limitations (lines 134/135). I agree and thus strongly recommend to include a table with the interpolation method/tool used for each variable, ideally with a justification for that choice. The devil lies in the detail and this will be important for reproducibility. At least I couldn't find such detailed information in the appendix (might have overlooked something).
- line 144: you mention the definition of the daily climatology here, averaged over 1981-2010. Daily values can still be very noisy (depending on the variable, but certainly for surface temperature) when averaged over such a short period. You might want to consider to run a Fourier decomposition, keeping the first few harmonics, to achieve even better results. See e.g. second to last paragraph in section 2.1 of the following paper: https://wcd.copernicus.org/articles/2/581/2021/
- I found the ABC methods hard to grasp without referring back to the original papers. For example, for Climatology++ you mention an improvement by additionally predicting the 'historical mean or geographic median over all days in a window around the target day of year'. I don't quite understand this operation. Is it effectively smoothing, too, and how does it actually differ from a climatology? Similarly, I find it hard to understand the other methods. What does it mean for persistence++ that a linear least squares regression is fit? To what data exactly? Fitted in space? In time?

**Relation To Prior Work:**

The links have been made appropriately.

**Summary And Contributions:**

This paper by Mouatadid et al. introduces a curated dataset of observations and dynamical forecast data to be used for training machine learning (ML) models for subseasonal weather prediction over the contiguous US. The paper is very well-written and runs through comprehensive benchmark experiments. There are a few suprising results, with simpler methods outperforming more complex ones, but this is a feature often found in weather and climate modelling.

Overall, the paper makes an important contribution in providing a benchmark dataset for future ML approaches, and supersedes the far more limited SubX Experiment.

Given the relatively good performance of the simpler methods, it would be particularly important to describe those methods as clearly as possible, without the readers having to refer back to the original data sources. My comments and suggestions therefore mainly refer to possible clarifications.

---

> ### Author Response · Authors · 2023-08-13
> **Climate adaptation**
>
> We apologize for the confusion and will clarify that by “climate adaptation” we mean enabling communities to adapt to the consequences of a changing climate by providing earlier advanced notice of atypical or problematic weather patterns.

---

> ### Author Response · Authors · 2023-08-13
> **Temporal resolution**
>
> Thank you for these valuable recommendations. We will highlight this information and add clarifications in the relevant sections as well as in Figure 1. Regarding homogenisation, all data from different sources goes through a pre-processing pipeline, resulting in standardized dataframes. As part of this pipeline, we standardize all data spatially by interpolating relevant input data to a 1ºx1º latitude-longitude grid and temporally by aggregating daily input values over a 14-day rolling window (lines 91-95). Daily precipitation data is aggregated using a sum (i.e., accumulated precipitation) while daily temperature is aggregated using an average (.i.e., mean temperature). Our data is averaged in a continuously rolling manner over 14 days (on our datasets, the column `start date` indicates the first day in this 14-day window), but it is oftentimes provided daily (i.e., every day of the year appears as a `start date`). Our evaluation data is comprised of the years 2011-2020 (cf. lines 203-204), but the SubseasonalClimateUSA includes much earlier data that can be used for model training (the first year available for each feature varies; e.g., 1979 for temperature and 1948 for precipitation; Appendix B includes the data used by each model for training purposes).

---

> ### Author Response · Authors · 2023-08-13
> **Advantages of merging observational and model data**
>
> Thank you for this recommendation.  In the final version, we will emphasize that both the present work and past studies have found complementary predictive signals in physics-based dynamical model forecasts and pure observational data that can lead to better forecasts than either data source alone [see, e.g., 22, 37].  In fact, recent work has demonstrated that even the least skillful SubX models can produce forecasts with skill comparable to the best when corrected suitably with observational data [37].  As a result, we endeavored to include both granular measurements and granular model forecasts in the SubseasonalClimateUSA dataset to best equip future model developers, researchers, and forecasters.

---

> ### Author Response · Authors · 2023-08-13
> **Interpolation**
>
> Thank you for this suggestion! Careful details regarding the interpolation methods and commands used can currently be found in the subseasonal_data GitHub repository  (https://github.com/microsoft/subseasonal_data/blob/main/DATA.md), and we will also add a table summarizing the relevant information to the final paper appendix.

---

> ### Author Response · Authors · 2023-08-13
> **Climatology**
>
> Thank you for this suggestion. It is worth clarifying two points concerning the climatology calculation: (1) we adopted the exact definition of climatology used by the US Bureau of Reclamation and NOAA in their Subseasonal Forecast Rodeo competitions [Nowak et al., 2020]; (2) while our climatology values are daily, they represent the average temperature over the 14 day period beginning on a given date (rather than average temperature for a single day).  This additional level of averaging, which is carried out in a rolling fashion, reduces the variability of the climatology values.  Nevertheless, we will investigate how the proposed Fourier decomposition approach can help further stabilize climatology in our dataset.
>
>
> ------
> References
>
>  K. Nowak, I.M Ferguson, J. Beardsley, and L. D Brekke. Enhancing western united states sub-seasonal forecasts: Forecast rodeo prize competition series. In AGU Fall Meeting 2020. AGU, 2020.

---

> ### Author Response · Authors · 2023-08-13
> **ABC**
>
> We apologize for the confusion and will clarify the following points in the revision. Climatology++ is indeed a form of smoothing, and it can be viewed as an adaptive form of climatology that learns how many prior years and how many dates in a window around the target date to include in a smoothed mean or geometric median estimate for a given grid point, target date, and target variable; importantly, unlike a static climatology, Climatology++ allows these learned window sizes to vary over time to adapt to noise levels and variability.  For each grid point and target date, Persistence++ predicts a target variable as a function of lagged measurements, climatology, and model forecasts observable for the same grid point on the forecast issuance date.  These features are combined using a linear least squares regression trained on all available historical data available as the forecast issuance date.  We will also add algorithm blocks for the ABC methods to the appendix to dispel future confusion and highlight that ABC is a hybrid physics+learning approach that takes as input a dynamical model forecast and uses the historical record of prior forecasts and observations to correct the model’s output and improve predictive skill.

---

### Author Response · Authors · 2023-08-13
**Thank you!**

We thank all reviewers for their detailed, positive, and constructive feedback.  We are especially pleased that reviewers found our submission to be a “strong contribution on an important topic,” well-written, comprehensive, and “highly valuable from the ethical and social implications standpoints.”  We reply to each reviewer’s questions and concerns below.

---

### Decision · Program_Chairs · 2023-09-22

**Decision:**

Accept (Poster)

**Comment:**

The paper introduces a dataset on subseasonal weather prediction utilizing observation data and physics-based dynamical models. The main contribution is aggregating data from diverse sources, formats, and time and spatial resolution. It is an important step for developing effective forecasting models. It is unclear how we evaluate the quality of the dataset - as mentioned in the review, the resolution could be too coarse for some important studies. The authors are encouraged to address the challenge in the final version.